# Data-driven Mixed Integer Optimization through Probabilistic Multi-variable Branching

Yanguang Chen [* 1]  Wenzhi Gao [* 2]  Wanyu Zhang [* 2]  Dongdong Ge [3 4]  Huikang Liu [3]  Yinyu Ye [3 4 2]

## Abstract

This paper introduces Probabilistic Multi-variable Branching (PMVB), a simple and effective technique for accelerating mixed-integer optimization using data-driven machine learning models. At its core, PMVB employs a multi-variable branching procedure that partitions the feasible region via data-driven hyperplanes and requires only two lines of code to implement. Moreover, PMVB is model-agnostic and compatible with a wide range of machine learning models. Leveraging tools from statistical learning theory, we develop interpretable hyperparameter selection strategies and propose several extensions to further enhance performance. We evaluate PMVB by integrating it into state-of-the-art MIP solvers and conducting experiments on both classical benchmark datasets and real-world instances. The results demonstrate the effectiveness of PMVB in improving MIP-solving efficiency.

## 1. Introduction

Mixed Integer Programming (MIP) (Sahinidis, 2019; Wolsey, 2020) is a core modeling tool in applications such as revenue management (Benati & Rizzi, 2007), production planning (Pochet & Wolsey, 2006), and portfolio optimization (Wolsey & Nemhauser, 1999). Despite its broad applicability, MIP is hard to solve, and state-of-the-art solvers (Gurobi Optimization, 2021; Ge et al., 2022) rely on heuristics to perform effectively in practice. Designing effective heuristics has been a long-standing topic in MIP solving.

Among various MIP applications, some exhibit a strong online nature: on a regular basis, similar models are solved with different data parameters. For example, in energy management (Wood et al., 2013) and production planning (Pochet & Wolsey, 2006), MIPs are typically solved daily. Although this online nature imposes restrictions on the solution time, it ensures the availability of offline datasets, which opens the way for data-driven machine learning approaches. In this aspect, many successful ML+MIP attempts (Bengio et al., 2021; Zhang et al., 2023; Han et al., 2023; Xavier et al., 2021; Alvarez et al., 2017; Liu et al., 2022) have been made and encouraging results have been reported. However, many successful ML+MIP attempts resort to complex modifications to the MIP solution process. Although these modifications often bring significant performance gains, they typically cannot be applied to commercial solvers without access to their source code. Additionally, machine learning methods are sometimes less interpretable, making hyperparameter tuning challenging.

In view of these challenges, we propose PMVB, a Probabilistic Multi-Variable Branching approach for MIP solving. PMVB relies on a probabilistic prediction of the binary (integer) variables and applies a simple multi-variable branching procedure that constructs branching hyperplanes

$$\sum_{i \in \mathcal{S}} y_i \leq z \quad \text{and} \quad \sum_{i \in \mathcal{S}} y_i \geq z + 1, \quad \{y_i\} \text{ binary},$$

to split the MIP feasible region into disjoint subregions. It can be implemented (in Python) in two lines of code:

```python
model.addConstr(sum(y[i] for i in S) <= z)
model.addConstr(sum(y[i] for i in S) >= z + 1)
```

**Contributions.**

- We propose PMVB, a method for leveraging offline datasets to accelerate online MIP solving. Our approach is simple yet efficient, model-agnostic, and compatible with existing machine learning models and MIP solvers.

- We apply statistical learning theory to justify PMVB and provide an interpretable framework for selecting its hyperparameters. We also develop several extensions of PMVB to enhance its practical performance, and comprehensive numerical experiments on various benchmark datasets demonstrate the effectiveness of our method.

---

[1] Shanghai University of Finance and Economics [2] Stanford University [3] Shanghai Jiao Tong University [4] SIMIS. Correspondence to: Wenzhi Gao <gwz@stanford.edu>.

*Proceedings of the 43rd International Conference on Machine Learning*, Seoul, South Korea. PMLR 306, 2026. Copyright 2026 by the author(s).

## 1.1. Related Work

**ML+MIP.** Machine learning has been applied in many aspects of MIP solving (Gasse et al., 2022b), and we focus on the attempts that have a direct relation with MIP solving. Alvarez et al. (2017); Khalil et al. (2016); Nair et al. (2020) use machine learning models to imitate the behavior of an expensive branching rule in MIP. Ding et al. (2020) propose a graph representation of MIP and predict the binary solution using a graph convolutional network. They also propose adding a cut that upper-bounds the distance between the prediction and the MIP solution, but do not further analyze this method. Nair et al. (2020) also adopt a graph representation of MIP to predict binary variables, and propose to combine the prediction with the MIP diving heuristic (neural diving). In (Gasse et al., 2019), the authors propose to learn the branching variable selection policy during the branch-and-bound procedure. Lee et al. (2019) adopt imitation learning to learn a good pruning policy for a nonlinear MIP application. Han et al. (2023) propose Predict-and-Search, a prediction-guided primal heuristic that restricts the MIP search to a trust region around the predicted binary solution. Other recent successful attempts in this direction include (Liu et al., 2025a; Cantürk et al., 2024; Turner et al., 2023; Du et al., 2025; Liu et al., 2025b). In addition, there has been growing interest in using reinforcement learning to accelerate MIP solving (Berthold & Hendel, 2021; Scavuzzo et al., 2022; Song et al., 2020; Wu et al., 2021; Zhang et al., 2024; Turner et al., 2025), and we refer the readers to (Bengio et al., 2021; Zhang et al., 2023; Wang et al., 2024) as well as Appendix **A** for a more detailed survey of recent `ML/RL+MIP` attempts.

**Multi-variable Branching.** In the MIP literature, multi-variable branching (MVB), also known as local branching, is an extension of single-variable branching (Karamanov & Cornuéjols, 2011; Fischetti & Lodi, 2003). Concretely, MVB generates so-called *branching hyperplanes* that partition the feasible region (say $\mathcal{Y}$) into disjoint subsets

$$\mathcal{Y} \cap \{\mathbf{y} : \langle \mathbf{a}, \mathbf{y} \rangle \leq z\} \quad \text{and} \quad \mathcal{Y} \cap \{\mathbf{y} : \langle \mathbf{a}, \mathbf{y} \rangle > z\}, \quad (1)$$

and the most widely used single-variable branching is a special case of (1), with vector $\mathbf{a}$ being a standard basis vector and $z = 0$. There is relatively sparse literature on MVB. Gamrath et al. (2015) conduct a systematic investigation of the performance of MVB, showcasing performance improvement on certain classes of MIP instances; Yang et al. (2021) perform a case study of MVB on knapsack problems and provide a theoretical analysis. The approach developed in this paper falls into the scope of MVB.

**Learning Theory for MIP.** Though most `MIP+ML` studies are empirical, there exist works in statistical learning theory that study the learnability of MIPs. In (Balcan et al., 2018), the authors derive the generalization bound for learning hybrid branching rules that take a convex combination of different score functions. In (Balcan et al., 2022a), the bound is improved for the MIP branch and bound procedure. In the same line of work, Balcan et al. (2022b) analyze the learnability of mixed integer and Gomory cutting planes. Recently, Khalife & Lodi (2025) showed a complexity lower bound for learning to cut.

**Outline of the Paper.** The paper is organized as follows. Section **2** sets up the problem and introduces the idea behind `PMVB`. We further provide an analysis of its behavior using statistical learning theory. In Section **3**, we discuss the practical aspects when implementing `PMVB`. Section **4** discusses the data-free variant of `PMVB` where we consider the root LP solution as an ML model output. Finally, Section **5** conducts experiments on both synthetic and real-life MIP instances to demonstrate the efficiency of `PMVB`.

## 2. Probabilistic Multi-variable Branching

This section presents the ideas behind `PMVB`. In short, `PMVB` constructs data-driven branching hyperplanes that partition the feasible region into disjoint subregions. These hyperplanes are derived from the probabilistic prediction of binary variables as well as concentration inequalities. We begin by setting up the background for parametric MIPs.

### 2.1. Parametric MIP and Solution Map

Consider a parametric MIP instance with $n$ binary variables and data parameter $\xi$, denoted by $\mathcal{P}(\xi)$ :

$$\mathcal{P}(\xi) := \min_{\mathbf{x} \in \mathbb{R}^d, \mathbf{y} \in \{0,1\}^n} \quad c(\mathbf{x}, \mathbf{y}; \xi) \qquad (2)$$
$$\text{subject to} \quad h(\mathbf{x}, \mathbf{y}; \xi) \leq 0.$$

In the formulation (2), the functions $c, h$ encapsulate the structural components of the problem, and $\xi \sim \Xi$ represents data parameters drawn from some distribution $\Xi$. Throughout, we assume the problem with fixed binary variables is tractable and focus on problems involving only binary variables, as is the case for mixed-integer linear programs. Let $\mathbf{y}^\star(\xi) = (y_1^\star(\xi), \ldots, y_n^\star(\xi))^\top \in \mathbb{R}^n$ denote the unique optimal solution to (2)[1].

*Remark* 2.1. We focus on binary variables for simplicity, but `PMVB` can be generalized to handle integer variables, for example, through binary representation of integers.

---

[1]We assume that the optimal solution map is unique throughout the paper. This assumption can often be satisfied by adding an arbitrarily small perturbation to the MIP objective.

## 2.2. Concentration Inequality and MVB

This section connects concentration inequalities in statistical learning with multi-variable branching in MIP. For ease of exposition, we leave a more rigorous background setup to Appendix **B**. With the setup in Section **2.1**, given a new MIP instance parameterized by $\xi$, suppose we have in hand a probabilistic prediction $\tilde{\mathbf{y}} = \tilde{\mathbf{y}}(\xi) \in [0,1]^n$. Informally, this prediction $\tilde{\mathbf{y}}$ can be interpreted as the likelihood, or the probability that each binary variable takes 1 in the optimal solution $\mathbf{y}^\star = \mathbf{y}^\star(\xi)$:

$$\tilde{y}_i = \mathbb{P}\{y_i^\star = 1\} = \mathbb{E}[y_i^\star].$$

In this sense, each $y_i^\star$ can be interpreted as a Bernoulli random variable Bernoulli$(\tilde{y}_i)$. To provide intuition, for now, we further make the restrictive assumption that these random variables $\{y_i^\star\}$ are independent. To leverage the prediction $\tilde{\mathbf{y}}$ in solving the instance $\mathcal{P}(\xi)$, consider the strategy of fixing each binary variable at the rounded solution $\hat{y}_i := \lfloor \tilde{y}_i + \frac{1}{2} \rfloor \in \{0,1\}$. Let $\mathcal{U}$ denote the set of binary variables fixed at the upper bound 1. Then the probability of fixing all such binary variables correctly is given by

$$\mathbb{P}\{\hat{y}_i = y_i^\star, \text{ for all } i \in \mathcal{U}\} = \prod_{i=1}^{|\mathcal{U}|} \tilde{y}_i. \tag{3}$$

As $|\mathcal{U}|$ increases, the chance of making correct fixing decisions drops exponentially, and fixing variables in $\mathcal{U}$ will very likely exclude the optimal solution $\mathbf{y}^\star$. In other words, the risks accumulate when each risky object (random variable) is treated separately. In view of this, it is natural to consider *risk pooling*: when the random variables are pooled, their internal noise will cancel each other out and allow for much safer decisions. In particular, for independent random variables, this idea of pooling can be explicitly characterized by concentration inequalities.

**Lemma 2.2.** *Given independent Bernoulli random variables $Y_1, \ldots, Y_n$ such that $\mathbb{P}\{Y_i = 1\} = \tilde{y}_i$, we have*

$$\mathbb{P}\{\big| \sum_{i=1}^n Y_i - \sum_{i=1}^n \tilde{y}_i \big| \geq t\} \leq 2 \cdot \exp(-\tfrac{2}{n}t^2)$$

*for any $t > 0$.*

Lemma **2.2** shows that when many Bernoulli random variables are aggregated together ($n$ being large), then the probability that their realization $\sum_{i=1}^n Y_i$ deviates from the expectation $\sum_{i=1}^n \tilde{y}_i$ by $t$ will vanish exponentially in $t^2$. It is in sharp contrast with the exponential risk accumulation observed in variable fixing (3). In particular, plugging in $Y_i = y_i^\star$ shows that the inequality

$$\mathcal{C}_{\mathcal{U}}: \quad \sum_{i \in \mathcal{U}} y_i^\star \geq \sum_{i \in \mathcal{U}} \tilde{y}_i - t_{\mathcal{U}}$$

holds with probability at least $1 - 2\exp(-\tfrac{2}{|\mathcal{U}|}t^2)$. Repeating the above argument for variables rounded to 0, denoted by $\mathcal{L}$, gives one more set of inequalities

$$\mathcal{C}_{\mathcal{L}}: \quad \sum_{i \in \mathcal{L}} y_i^\star \leq \sum_{i \in \mathcal{L}} \tilde{y}_i + t_{\mathcal{L}}.$$

---

**Algorithm 1** Probabilistic Multi-variable Branching

**input** Prediction $\tilde{\mathbf{y}}$ and rounded solution $\hat{\mathbf{y}}$
1: Let $\mathcal{U} = \{i : \hat{y}_i = 1\}$ and $\mathcal{L} = \{i : \hat{y}_i = 0\}$
2: Generate $\mathcal{C}_{\mathcal{U}}, \mathcal{C}_{\mathcal{L}}, \bar{\mathcal{C}}_{\mathcal{U}}, \bar{\mathcal{C}}_{\mathcal{L}}$ and four subproblems

$$\{\mathcal{P}_{\mathcal{C}_{\mathcal{U}}, \mathcal{C}_{\mathcal{L}}}, \mathcal{P}_{\mathcal{C}_{\mathcal{U}}, \bar{\mathcal{C}}_{\mathcal{L}}}, \mathcal{P}_{\bar{\mathcal{C}}_{\mathcal{U}}, \mathcal{C}_{\mathcal{L}}}, \mathcal{P}_{\bar{\mathcal{C}}_{\mathcal{U}}, \bar{\mathcal{C}}_{\mathcal{L}}}\}$$

3: Solve $\mathcal{P}_{\mathcal{C}_{\mathcal{U}}, \mathcal{C}_{\mathcal{L}}}$ and get the optimal solution $\mathbf{y}^\star_{\mathcal{C}_{\mathcal{U}}, \mathcal{C}_{\mathcal{L}}}$
4: (Optionally) Solve $\mathcal{P}_{\mathcal{C}_{\mathcal{U}}, \bar{\mathcal{C}}_{\mathcal{L}}}, \mathcal{P}_{\bar{\mathcal{C}}_{\mathcal{U}}, \mathcal{C}_{\mathcal{L}}}, \mathcal{P}_{\bar{\mathcal{C}}_{\mathcal{U}}, \bar{\mathcal{C}}_{\mathcal{L}}}$ and get

$$\{\mathbf{y}^\star_{\mathcal{C}_{\mathcal{U}}, \bar{\mathcal{C}}_{\mathcal{L}}}, \mathbf{y}^\star_{\bar{\mathcal{C}}_{\mathcal{U}}, \mathcal{C}_{\mathcal{L}}}, \mathbf{y}^\star_{\bar{\mathcal{C}}_{\mathcal{U}}, \bar{\mathcal{C}}_{\mathcal{L}}}\}$$

**output** best among $\{\mathbf{y}^\star_{\mathcal{C}_{\mathcal{U}}, \mathcal{C}_{\mathcal{L}}}, \mathbf{y}^\star_{\mathcal{C}_{\mathcal{U}}, \bar{\mathcal{C}}_{\mathcal{L}}}, \mathbf{y}^\star_{\bar{\mathcal{C}}_{\mathcal{U}}, \mathcal{C}_{\mathcal{L}}}, \mathbf{y}^\star_{\bar{\mathcal{C}}_{\mathcal{U}}, \bar{\mathcal{C}}_{\mathcal{L}}}\}$

---

The inequalities $\mathcal{C}_{\mathcal{U}}, \mathcal{C}_{\mathcal{L}}$ partition the MIP feasible region into four subregions $\mathcal{P}_{\mathcal{C}_{\mathcal{U}}, \mathcal{C}_{\mathcal{L}}}, \mathcal{P}_{\bar{\mathcal{C}}_{\mathcal{U}}, \mathcal{C}_{\mathcal{L}}}, \mathcal{P}_{\mathcal{C}_{\mathcal{U}}, \bar{\mathcal{C}}_{\mathcal{L}}}, \mathcal{P}_{\bar{\mathcal{C}}_{\mathcal{U}}, \bar{\mathcal{C}}_{\mathcal{L}}}$, with $\mathcal{P}_{\mathcal{C}_{\mathcal{U}}, \mathcal{C}_{\mathcal{L}}}$ containing the optimal solution with high probability. Given that $\mathcal{P}_{\mathcal{C}_{\mathcal{U}}, \mathcal{C}_{\mathcal{L}}}$ likely contains the optimal solution $\mathbf{y}^\star$, the idea of PMVB (Algorithm **1**) is straightforward:

- either focus on solving $\mathcal{P}_{\mathcal{C}_{\mathcal{U}}, \mathcal{C}_{\mathcal{L}}}$ and ignore the rest,

- or treat $\mathcal{C}_{\mathcal{U}}, \mathcal{C}_{\mathcal{L}}$ as branching hyperplanes and solve four subproblems to ensure exactly solving $\mathcal{P}(\xi)$.

In the first case, PMVB serves as a primal heuristic, while in the second, PMVB becomes an exact multi-variable branching strategy that incorporates data-driven information. In either case, PMVB only relies on a black-box probabilistic prediction $\tilde{\mathbf{y}}$ and is compatible with any machine learning model that provides such a probabilistic prediction.

Before concluding the section, we note that the only hyperparameter in PMVB is the intercept of the hyperplane

$$\sum_{i \in \mathcal{U}} \tilde{y}_i - t_{\mathcal{U}} \quad \text{and} \quad \sum_{i \in \mathcal{L}} \tilde{y}_i + t_{\mathcal{L}},$$

which can be set to $\sqrt{\frac{1}{2|\mathcal{U}|} \log(\frac{4}{\delta})}$ and $\sqrt{\frac{1}{2|\mathcal{L}|} \log(\frac{4}{\delta})}$ to achieve a success probability of $1 - \delta$ in the concentration inequalities from Lemma **2.2**.

Although the settings in this section suffice to demonstrate the main idea of PMVB and for it to be used heuristically, the analysis so far imposes several restrictions. First of all, the independence assumption for Lemma **2.2** is too strong and almost never holds in practice. While we show in Appendix **B** a more rigorous learning-theory-based derivation of PMVB, it still relies on restrictive assumptions. In addition, the current choice $\mathcal{U}$ and $\mathcal{L}$ involve all the binary variables, which may not be sensible when the probabilistic prediction $\tilde{\mathbf{y}}$ contains values near $\frac{1}{2}$. Finally, when PMVB is employed to solve the MIP exactly, it is unclear whether solving four subproblems would actually be faster than solving the original problem. Section **3** addresses the above issues by discussing the practical aspects of PMVB.

# 3. Practical Aspects of PMVB

Building on the intuitions from Section **2**, this section explores practical considerations to further improve PMVB.

## 3.1. Practical PMVB without Independence Assumption

In practice, a probabilistic prediction $\tilde{\mathbf{y}}$ can contain values near $\frac{1}{2}$, which correspond to binary variables with low prediction confidence. Including these low-confidence variables weakens the strength of the branching hyperplanes. Instead of adding all the variables, set some threshold $\tau \in (\frac{1}{2}, 1]$ and define the (threshold-)rounded solution:

$$\hat{y}_i(\xi) = \begin{cases} 1 & \text{if } \tilde{y}_i(\xi) \geq \tau, \\ 0 & \text{if } \tilde{y}_i(\xi) \leq 1 - \tau, \\ \text{not rounded} & \text{otherwise.} \end{cases} \quad (4)$$

For any fixed data parameter $\xi$ and threshold $\tau$, define the (thresholded) index sets of variables predicted at their lower and upper bounds as:

$$\mathcal{L}(\tau; \xi) := \{i : \tilde{y}_i(\xi) \leq 1 - \tau\}, \quad (5)$$
$$\mathcal{U}(\tau; \xi) := \{i : \tilde{y}_i(\xi) \geq \tau\}. \quad (6)$$

Using these sets, we define two measures of prediction accuracy as follows:

$$\alpha_{\mathcal{L}}(\tau; \xi) = \frac{1}{|\mathcal{L}(\tau;\xi)|} \sum_{i \in \mathcal{L}(\tau;\xi)} \mathbb{I}\{y_i^\star(\xi) = 0\}, \quad (7)$$
$$\alpha_{\mathcal{U}}(\tau; \xi) = \frac{1}{|\mathcal{U}(\tau;\xi)|} \sum_{i \in \mathcal{U}(\tau;\xi)} \mathbb{I}\{y_i^\star(\xi) = 1\}, \quad (8)$$

where $\mathbb{I}\{\cdot\}$ denotes the 0-1 indicator function. The quantities $\alpha_{\mathcal{L}}(\tau; \xi)$ and $\alpha_{\mathcal{U}}(\tau; \xi)$ quantify the prediction accuracy of the model $\tilde{y}$ for threshold $\tau$ on any given instance $\xi$. Different choices of the hyperparameter $\tau$ can result in varying prediction accuracies, thereby influencing the performance of PMVB. Figure **1** illustrates the mean and variance of $\alpha_{\mathcal{L}}(\tau; \xi)$ and $\alpha_{\mathcal{U}}(\tau; \xi)$ over $\xi \in \Xi$, as well as the average sizes of the index sets $\mathcal{L}(\tau; \xi)$ and $\mathcal{U}(\tau; \xi)$, as functions of the threshold parameter $\tau$. These statistics are computed using predictions $\{\tilde{y}_i(\xi)\}$ obtained from a GNN model, based on 100 independent set (IS-V) instances (see Section **5** for more details).

As shown in Figure **1**, there is a trade-off with respect to the choice of $\tau$. A larger $\tau$ tends to yield higher prediction accuracy and lower variance but includes fewer variables, potentially resulting in weaker hyperplanes. Conversely, a smaller $\tau$ includes more variables but leads to lower accuracy and higher variance. Thus, we recommend choosing the threshold $\tau$ according to:

$$\tau^\star = \max_{\tau \in (\frac{1}{2}, 1]} \{\tau : \mathbb{E}[\alpha_{\mathcal{L}}(\tau; \xi)] \geq \tau, \mathbb{E}[\alpha_{\mathcal{U}}(\tau; \xi)] \geq \tau\}. \quad (9)$$

The intuition behind the selection rule (9) is to strike a balance between quantity (the number of variables included)

and quality (the accuracy of predictions). Similar principles for balancing prediction quality and model complexity can be found in statistical model selection methods, such as Mallows' $C_p$ criterion (Gilmour, 1996), $k$-fold cross-validation (Bengio & Grandvalet, 2004), and related techniques in reliability theory and extreme value theory.

Let $\sigma^2$ denote an upper bound on the variance of both accuracy metrics $\alpha_{\mathcal{L}}(\tau; \xi)$ and $\alpha_{\mathcal{U}}(\tau; \xi)$, that is,

$$\mathbb{V}[\alpha_{\mathcal{L}}(\tau; \xi)] \leq \sigma^2 \quad \text{and} \quad \mathbb{V}[\alpha_{\mathcal{U}}(\tau; \xi)] \leq \sigma^2. \quad (10)$$

By selecting the threshold $\tau$ according to the rule (9) and applying Chebyshev's inequality (see Lemma **C.3** in Appendix **C**) for any $\delta \in (0, 1)$, we obtain a probabilistic guarantee for prediction accuracy:

$$\mathbb{P}\{|\alpha_{\mathcal{U}}(\tau; \xi) - \mathbb{E}[\alpha_{\mathcal{U}}(\tau; \xi)]| \leq \tfrac{\sigma}{\sqrt{\delta}}\} \geq 1 - \delta, \quad (11)$$

and a similar inequality holds for $\alpha_{\mathcal{L}}(\tau; \xi)$. Thus, combining (7), (9), and (11) together, we have the following theorem, which is a more practical version of Lemma **2.2**.

**Theorem 3.1.** *Let $\mathcal{U} = \mathcal{U}(\tau; \xi)$ and $\mathcal{L} = \mathcal{L}(\tau; \xi)$ be defined as in* (5)*, with $\tau$ chosen according to* (9)*, and suppose $\sigma$ satisfies* (10)*. Then we have*

$$\mathbb{P}\{\textstyle\sum_{i \in \mathcal{U}} y_i^\star(\xi) \geq \tau|\mathcal{U}| - \tfrac{\sigma|\mathcal{U}|}{\sqrt{\delta}}\} \geq 1 - \delta,$$
$$\mathbb{P}\{\textstyle\sum_{i \in \mathcal{L}} y_i^\star(\xi) \leq (1-\tau)|\mathcal{L}| + \tfrac{\sigma|\mathcal{L}|}{\sqrt{\delta}}\} \geq 1 - \delta.$$

*for any $\delta \in (0, 1)$.*

Since Theorem **3.1** follows directly from Chebyshev's inequality, its proof is omitted. It is worth noting that, compared to the argument in Section **2.2**, Theorem **3.1** does not rely on any additional assumptions, making it more broadly applicable in practice. However, this generality comes at a cost: without those assumptions, the probability bound in Theorem **3.1** no longer decays exponentially and is therefore more conservative.

*Remark* 3.2. Consider an example where the mean prediction accuracy is illustrated in Figure **1**. Suppose the threshold $\tau = 0.9$, the confidence parameter $\delta = 0.05$, and estimate the variance bound $\sigma \approx 0.025$ according to Figure **1**. If $\mathcal{U}(\tau; \xi) = \{1, 2, \ldots, 100\}$, then the hyperplane becomes

$$\textstyle\sum_{i=1}^{100} y_i^\star(\xi) \geq 90 - 100 \cdot \tfrac{0.025}{\sqrt{0.05}} \geq 78.$$

That is, with probability no less than 95%, at least 78 out of the 100 variables take 1.

*Remark* 3.3. In practice, according to the definition of $\mathcal{U}(\tau; \xi)$ and $\mathcal{L}(\tau; \xi)$, we have $\sum_{i \in \mathcal{U}(\tau;\xi)} \tilde{y}_i(\xi) \geq \tau|\mathcal{U}(\tau;\xi)|$ and $\sum_{i \in \mathcal{L}(\tau;\xi)} \tilde{y}_i(\xi) \leq (1-\tau)|\mathcal{L}(\tau;\xi)|$. Hence, it is possible to leverage the following two inequalities to get further tightened branching hyperplanes:

$$\mathcal{C}'_{\mathcal{U}} : \textstyle\sum_{i \in \mathcal{U}(\tau;\xi)} y_i(\xi) \geq \sum_{i \in \mathcal{U}(\tau;\xi)} \tilde{y}_i(\xi) - \tfrac{\sigma}{\sqrt{\delta}}|\mathcal{U}(\tau;\xi)|,$$
$$\mathcal{C}'_{\mathcal{L}} : \textstyle\sum_{i \in \mathcal{L}(\tau;\xi)} y_i(\xi) \leq \sum_{i \in \mathcal{L}(\tau;\xi)} \tilde{y}_i(\xi) + \tfrac{\sigma}{\sqrt{\delta}}|\mathcal{L}(\tau;\xi)|.$$

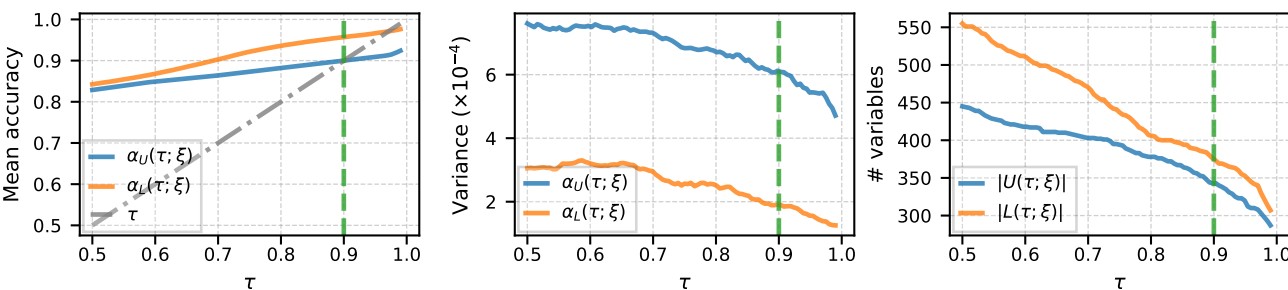

*Figure 1.* The mean and variance of $\alpha_{\mathcal{L}}(\tau;\xi)$ and $\alpha_{\mathcal{U}}(\tau;\xi)$, along with the average sizes of $\mathcal{L}(\tau;\xi)$ and $\mathcal{U}(\tau;\xi)$, computed over 100 independent set instances for varying values of the threshold parameter $\tau$.

### 3.2. Efficient Pruning using Objective Cuts

When PMVB is applied to solve MIP exactly as a branching rule (Algorithm **1**), it is necessary to solve all four subproblems to certify optimality. However, prior knowledge that the optimal solution is more likely to lie in $\mathcal{P}_{\mathcal{C}_{\mathcal{U}}, \mathcal{C}_{\mathcal{L}}}$ can be utilized to accelerate the solution procedure. Denote by $c^{\star}_{\mathcal{U},\mathcal{L}}$ to be the optimal value of $\mathcal{P}_{\mathcal{C}_{\mathcal{U}}, \mathcal{C}_{\mathcal{L}}}$. We introduce a small perturbation $\varepsilon > 0$ and enforce the additional constraint (objective cutting plane)

$$c(\mathbf{x}, \mathbf{y}; \xi) \leq c^{\star}_{\mathcal{U},\mathcal{L}} - \varepsilon \quad (12)$$

while solving the other subproblems $\mathcal{P}_{\bar{\mathcal{C}}_{\mathcal{U}}, \mathcal{C}_{\mathcal{L}}}, \mathcal{P}_{\mathcal{C}_{\mathcal{U}}, \bar{\mathcal{C}}_{\mathcal{L}}}$, and $\mathcal{P}_{\bar{\mathcal{C}}_{\mathcal{U}}, \bar{\mathcal{C}}_{\mathcal{L}}}$. Once the region $\mathcal{P}_{\mathcal{C}_{\mathcal{U}}, \mathcal{C}_{\mathcal{L}}}$ contains an optimal solution to $\mathcal{P}(\xi)$, the remaining three branching subproblems become infeasible under (12) and can often be pruned efficiently, at the cost of solving only three root relaxations.

## 4. Data-free PMVB using LP Relaxation

Despite being a data-driven method, the idea behind PMVB is applicable so long as a likelihood prediction of the binary variables is available. In the MIP context, a natural surrogate is the optimal solution to the root LP relaxation. Let $\mathbf{y}^{\star}_{\mathrm{Root}}$ be the MIP root relaxation solution; the idea is to construct hyperplanes based on $\mathbf{y}^{\star}_{\mathrm{Root}}$ instead. In other words, we view "LP solving" itself as a prediction algorithm. We justify the idea by analyzing the theoretical properties of data-free PMVB applied to the knapsack problem.

### 4.1. Analysis of Data-free PMVB

Consider the following knapsack problem:

$$\max_{\mathbf{y} \in \{0,1\}^n} \quad c_1 y_1 + \cdots + c_n y_n$$
$$\text{subject to} \quad a_1 y_1 + \cdots + a_n y_n \leq b, \quad (13)$$

where $a_i, c_i \geq 0$ and $b > 0$ are the data coefficients of the problem. We make the following assumptions:

**A1:** (Uniform cost) Parameters $\{a_i\}$ are i.i.d. uniformly distributed in $(0,1)$, i.e., $a_i \sim U[0,1]$.

**A2:** (Uniform ratio) The profit $c_i = f_i \cdot a_i$ where $\{f_i\}$ are i.i.d. uniformly distributed in $[0,1]$, i.e., $f_i \sim U[0,1]$.

Similar assumptions have also been made in (Lueker, 1982) to analyze the average difference of objective value between the 0/1 knapsack problem and its linear relaxation. There are many other works on both the average and worst-case performance of LP relaxation for knapsack problems, such as (Kohli et al., 2004; Morales & Martínez, 2020).

**Theorem 4.1.** *Let $\mathbf{y}^{\star}$ and $\tilde{\mathbf{y}}$ be the optimal solution to the knapsack problem* (13) *and its LP relaxation, respectively. Under* **A1** *and* **A2** *and assuming $b = \gamma n$ for some constant $\gamma \in (0, \frac{1}{2})$. Further let $\mathcal{U} := \{i : \tilde{y}_i = 1\}$ and $\mathcal{L} = \{i : \tilde{y}_i = 0\}$. Then each of the inequalities below holds*

$$\mathcal{C}_{\mathcal{U}} : \sum_{i \in \mathcal{U}} y^{\star}_i \geq \sum_{i \in \mathcal{U}} \tilde{y}_i - 4\sqrt{2}n^{\frac{3}{4}} = |\mathcal{U}| - 4\sqrt{2}n^{\frac{3}{4}},$$

$$\mathcal{C}_{\mathcal{L}} : \sum_{i \in \mathcal{L}} y^{\star}_i \leq \sum_{i \in \mathcal{L}} \tilde{y}_i + 4\sqrt{2}n^{\frac{3}{4}} + n^{\frac{1}{4}} = 4\sqrt{2}n^{\frac{3}{4}} + n^{\frac{1}{4}}.$$

*with probability at least $1 - \lceil 2\sqrt{n} \rceil \exp(\frac{-\sqrt{n}}{8})$.*

Theorem **4.1** shows that, for the 0/1-knapsack problem in the uniform distribution setting, branching hyperplanes generated from the LP solution hold with high probability. It is easy to see that $|\mathcal{U}| \geq b = \gamma n$, so that both $4\sqrt{2}n^{\frac{3}{4}}$ and $4\sqrt{2}n^{\frac{3}{4}} + n^{\frac{1}{4}}$ are $o(|\mathcal{U}|)$ with respect to the size of $\mathcal{U}$.

*Remark* 4.2. Our data-free argument also extends to multi-knapsack problem.

$$\max_{\mathbf{y} \in \{0,1\}^n} \quad \sum_{i=1}^{n} c_i y_i \quad \text{subject to} \quad \sum_{i=1}^{n} \mathbf{a}_i y_i \leq \mathbf{b},$$

where $\mathbf{a}_i \in \mathbb{R}^m_+$, $i = 1, \ldots, n$, and $\mathbf{b} \in \mathbb{R}^m_+$ are given vectors. Appendix **D.4** shows how the same reduced-cost argument can be extended to this multi-constraint setting.

### 4.2. Using Interior Point Solution in PMVB

In practice, it is observed that the branching hyperplanes generated from the interior point solution (from the interior point method) are often more effective than those derived from the vertex solution (from the simplex method). The simplex method always yields a vertex solution where most

binary variables take exactly binary values, while the interior point method gives fractional values more accurately reflecting the likelihood of the variables taking 0/1. Similar advantages of the interior point solution for MIP have also been observed in the literature (Berthold et al., 2018).

## 5. Numerical Experiments

In this section, we present experiments to evaluate the practical performance of different variants of `PMVB`, available on Github[2]. For the data-driven experiments, we use two commercial solvers: `Gurobi` (Gurobi Optimization, 2021) and `COPT` (Ge et al., 2022). Due to license restrictions, only `COPT` is used for the MIPLIB experiments. All experiments are conducted on a machine with an Intel(R) Xeon(R) CPU E5-2680 @ 2.70GHz and 64GB of memory.

### 5.1. Experiment Setup

**Machine Learning Model.** We adopt two types of machine learning models: simple logistic regression and graph neural network. Logistic regression is applied to tasks where only part of the coefficients in MIP change, and it has a low model complexity. We use a separate regression model for each binary variable and train $\hat{y}_i$ over the whole training set. The probabilistic output is used for generating branching hyperplanes. GNNs are used for more complex scenarios in which the problem size and data coefficients vary. We reuse the GNN models from (Cantürk et al., 2024), which is trained on 1000 instances and validated on 200 instances from each class. Appendix **E.2** further compares different GNN architectures and shows that higher prediction accuracy often leads to greater speedup.

**MIP Data for Logistic Regression.** We consider three test problem classes with fixed problem size for logistic regression: **1)**. multi-knapsack problem (`MKP`). **2)**. set covering problem (`SCP`) and **3)**. a real-world security-constrained unit commitment problem (`SCUC`) from `IEEE 118-bus system` as our testing models. The detailed testing data generation procedure is given below:

- For `MKP` models $\max_{\mathbf{Ay} \leq \mathbf{b}, \mathbf{y} \in \{0,1\}^n} \langle \mathbf{c}, \mathbf{y} \rangle$, $\mathbf{A} \in \mathbb{R}^{m \times n}$, we generate synthetic datasets according to the problem setup in (Chu & Beasley, 1998). After choosing problem sizes $(m, n) \in \{(10, 250), (10, 500), (30, 250), (30, 500)\}$, each element of $a_{ij}$ is uniformly drawn from $\{1, \ldots, 1000\}$ and $c_i = \frac{1}{m} \sum_j a_{ij} + \delta_i$ where $\delta_i$ is sampled from $\{1, \ldots, 500\}$. Then for each setup we fix $\mathbf{A}, \mathbf{c}$ and generate 500 i.i.d. instances with $b_i \sim \mathcal{U}[0.8 \cdot \frac{1}{4n} \sum_j a_{ij}, 1.2 \cdot \frac{1}{4n} \sum_j a_{ij}]$ for training. We

generate 20 new instances for testing. In `MKP` $\xi = \mathbf{b}$.

- For `SCP` models $\min_{\mathbf{Ay} \geq \mathbf{1}, \mathbf{y} \in \{0,1\}^n} \langle \mathbf{c}, \mathbf{y} \rangle$, $\mathbf{A} \in \mathbb{R}^{m \times n}$, we use dataset from (Umetani, 2017). After choosing problem sizes $(m, n) \in \{(1000, 10000), (1000, 20000), (2000, 20000)\}$ and generating $\mathbf{A}$ with sparsity 0.01, $\bar{c}_i \in \{1, \ldots, 100\}$, for each setup we fix $\mathbf{A}$ and generate 500 i.i.d. instances with $c_i \sim \mathcal{U}[0.8 \cdot \bar{c}_i, 1.2 \cdot \bar{c}_i]$ for training. We generate 20 new test instances. In `SCP` $\xi = \mathbf{c}$.

- For `SCUC` models $\min_{(\mathbf{x}, \mathbf{y}), h(\mathbf{x}, \mathbf{y}) \leq \mathbf{0}, \mathbf{Dx} = \mathbf{d}} c(\mathbf{x}, \mathbf{y})$, we use `IEEE 118` dataset (Birchfield et al., 2016; Xu et al., 2017) of time horizon 96, fix all generator parameters and simulate load demand vector $\mathbf{d} \in \mathcal{U}[l, u]^{96}$, $(l, u) \in \{(3000, 5000), (4000, 6000)\}$. For each setup, we generate 200 i.i.d. instances for training and 5 new instances for testing. In `SCUC` $\xi = \mathbf{d}$.

**MIP Data for GNN.** We consider three types of combinatorial problems of varying sizes for GNN: **1)** set covering (with varying sizes) (`SCP-V`) **2)** combinatorial auction (`CA-V`) **3)** independent set (`IS-V`). These datasets are taken from (Gasse et al., 2019). `SCP-V` has problem size with $(m, n) \approx (500, 1000)$, `CA-V` has problem size with $(m, n) \approx (400, 1000)$, and `IS-V` has problem size with $(m, n) \approx (8400, 2000)$. We refer the reader to Gasse et al. (2019) for the detailed formulations of `CA-V` and `IS-V`.

**MIPLIB Data.** Aside from the datasets above, we also use MIPLIB 2017 collection (Gleixner et al., 2021) to test data-free `PMVB`. We take 132 instances from the MIPLIB benchmark set (Gleixner et al., 2021).

**Training Data.** In the logistic regression tasks, we generated the training data ourselves: each `MKP` is given 300 seconds and solved to a $10^{-4}$ relative gap; Each `SCP` of size $(m, n) = (1000, 10000)$ is given 300 seconds and solved to $10^{-3}$ gap; Each `SCP` of size $(m, n) \in \{(1000, 20000), (2000, 20000)\}$ is given 600 seconds and solved to a $10^{-1}$ gap; Each `SCUC` is given 3600 seconds and solved to a $10^{-4}$ gap. For GNN, 1000 instances are generated according to the setup in (Cantürk et al., 2024; Gasse et al., 2019; Gleixner et al., 2021). The labels used for training are generated using `CPLEX`.

### 5.2. `PMVB` as a Primal Heuristic

In this section, we benchmark the performance of `PMVB` as a primal heuristic. We choose two leading MIP solvers `Gurobi` (Gurobi Optimization, 2021) and `COPT` (Ge et al., 2022) from the MIP benchmark (Mittelmann, 2020) and use them to evaluate performance improvement using `PMVB` to improve the primal upper bound.

Given a new MIP instance from our test set, we construct

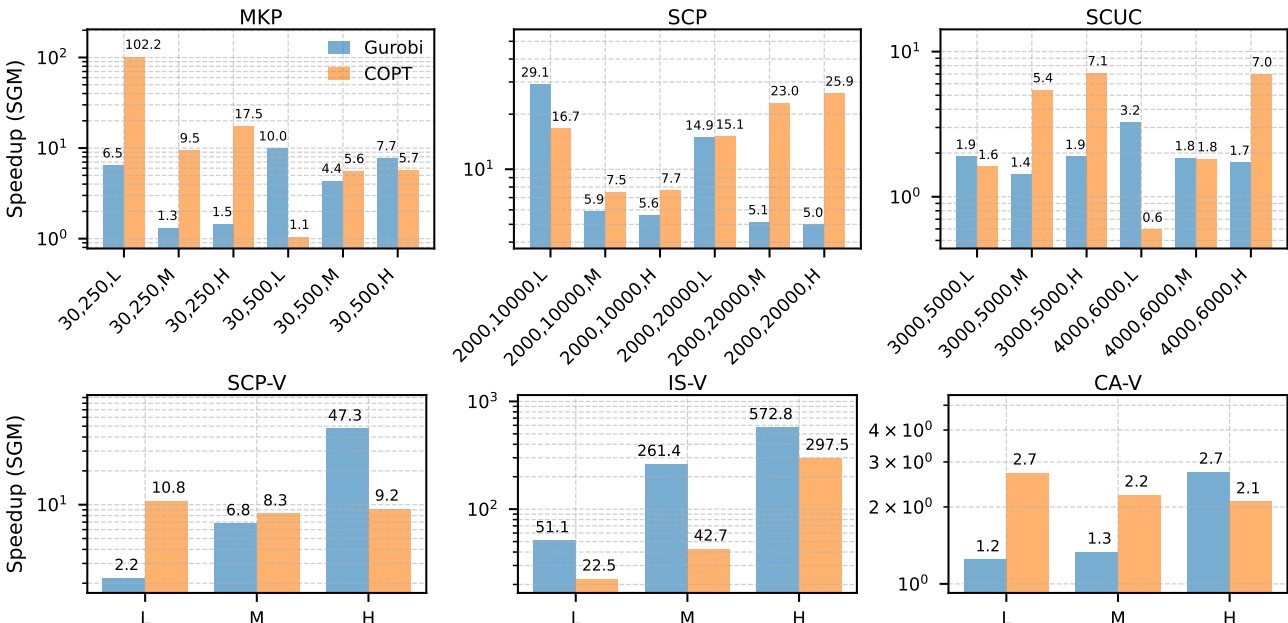

*Figure 2.* First row: logistic regression experiments on multi-knapsack (`MKP`), set-covering (`SCP`), and security-constrained unit-commitment (`SCUC`). Each group on the x-axis is a setting $(m, n, \text{Heuristic})$ and the y-axis is the average speedup $\Sigma$. For `MKP` and `SCP`, we show six settings with the largest problem size; the rest are reported in Figure **4** in the appendix.

Second row: GNN experiments on set covering `SCP-V`, independent set (`IS-V`), and combinatorial auction (`CA-V`). Recall that the suffix `-V` means the instances can have varying sizes. Each group on the x-axis corresponds to a Heuristic setting, and the y-axis shows the average speedup. The size of problems in each set is the same as in (Cantürk et al., 2024; Gasse et al., 2019).

the two hyperplanes according to Theorem **3.1** we set $\delta = 0.8, \tau = 0.9$ in this evaluation. After generating the hyperplanes, we allow 120 seconds for the solver to solve $\mathcal{P}_{\mathcal{C}_U, \mathcal{C}_L}$ and obtain the time $T_{\text{PMVB}}$ it spends achieving the best objective value $F_{\text{PMVB}}$. Then we run the solver again, with a time limit of 3600 seconds, on the MIP instance without adding hyperplanes, and record the time $T_{\text{Orig}}$ for the solver to reach $F_{\text{PMVB}}$. The solver is warm-started with the rounded prediction. We take the shifted geometric mean (SGM-10) (Mittelmann, 2020) over testing instances

$$\bar{T}_{\text{PMVB}} := \exp(\tfrac{1}{n} \textstyle\sum_{i=1}^n \log \max\{1, T_{\text{PMVB}}^i + 10\}) - 10,$$
$$\bar{T}_{\text{Orig}} := \exp(\tfrac{1}{n} \textstyle\sum_{i=1}^n \log \max\{1, T_{\text{Orig}}^i + 10\}) - 10.$$

We report the average speedup $\Sigma := \frac{\bar{T}_{\text{Orig}}}{\bar{T}_{\text{PMVB}}}$ as the primary performance evaluation metric. To assess how our method interacts with the built-in heuristics of MIP solvers, we vary these parameters across different levels. Specifically, we use L, M, and H to denote low, medium, and high heuristic settings, respectively. For `Gurobi`, these correspond to `Heuristics` values $\{0, 0.05, 1.0\}$; for `COPT`, they correspond to `HeurLevel` values $\{0, -1, 3\}$. Here, L means heuristics are disabled; M means the default setting, and H means applying the heuristics aggressively. To isolate the effect of `PMVB` from solver initialization, Appendix **E.4** reports the same evaluation without solver warm start and shows that the acceleration effect persists.

As shown in our experiments Figure **2**, when the problem structure remains fixed, even a simple logistic regression model enables `PMVB` to deliver substantial speedups on two commercial solvers across both synthetic datasets and real-world problems. For `MKP` instances, we observe up to a 5x speedup across several configurations. On the `SCUC` instances, `PMVB` still achieves an average speedup of over 50%. Notably, for `SCP` instances, we observe consistent speedups across all test cases, with several cases showing 10x speedup. These results highlight the practical effectiveness and robustness of our method.

When the problem structure is allowed to vary, the second row of Figure **2** demonstrates that the GNN model is capable of capturing structural patterns in the problem instances. However, the interaction between `PMVB` and the solvers' internal heuristics becomes less predictable across different datasets. For example, in the case of `SCP`, we observe a clear trend of diminishing speedup for both solvers as the problem becomes relatively easy (i.e., with small $(m, n)$). Conversely, as the problem size increases (e.g., $(m, n) = (2000, 20000)$), the performance benefits of `PMVB` become more evident.

Another observation is the solver-specific performance variation introduced by `PMVB`. While `COPT` generally benefits more than `Gurobi` in the logistic regression setting (Figure **2**, first row), the trend reverses when GNN is used

for prediction (Figure **2**, second row), where `Gurobi` sees greater improvements.

*Remark* 5.1. We also compare `PMVB` with Predict-and-Search (P&S) (Han et al., 2023), a primal heuristic that solves the MIP within a trust region around the predicted binary solution. Although `PMVB` and P&S both use probabilistic predictions, `PMVB` separately aggregates the sets of predicted zeros and ones into two hyperplanes, whereas P&S mixes them in a single trust-region constraint. We also conduct ablation studies to compare both methods in Appendix **E.3**

### 5.3. `PMVB` as a Branching Rule

In this section, we evaluate `PMVB` as an external branching rule integrated with `Gurobi`. We implement the objective cut pruning strategy described in Section **3.2** and test the method on the following datasets:

- 40 `MKP` instances using logistic regression;

- 200 `SCP-V` and `CA-V` instances using GNN.

All experiments are conducted using `Gurobi` with 4 threads and a time limit of 3600 seconds. All other solver parameters are kept at their default settings. The results are summarized in Table **1**.

*Table 1.* Comparison of original SGM and `PMVB` time. The suffix `-V` denotes problems with varying sizes across instances.

| Instance | $\bar{T}_{\texttt{Orig}}$ | $\bar{T}_{\texttt{PMVB}}$ | $\Sigma$ |
|----------|------|------|------|
| `SCP-V` | 32.6 | 27.4 | 1.18 |
| `CA-V` | 8.4 | 7.0 | 1.20 |
| `MKP` $(m,n) = (10, 500)$ | 28.7 | 25.8 | 1.11 |
| `MKP` $(m,n) = (30, 500)$ | 1283 | 979 | 1.31 |

According to Table **1**, we observe a 10% to 30% improvement in solution time. Moreover, as we discussed in Section **3.2**, the three subproblems are pruned within one second, and they are almost "free" to solve. This experiment reveals the efficiency of `PMVB` even when the underlying solver is only accessible as a black-box solver. The solution time improvement is not as impressive as when we employ `PMVB` as a primal heuristic. However, we believe a 20% improvement over the state-of-the-art solvers is still non-trivial, especially given the flexibility of our approach.

### 5.4. Data-free `PMVB`

Next, we evaluate the performance of `PMVB` as a data-free primal heuristic for MIP. In this setting, we use the interior-point solution of the LP relaxation as a surrogate for binary prediction. Appendix **E.5** further investigates the reliability of LP relaxation as a proxy on MIPLIB instances. `COPT` is used to solve the resulting problems, with a time limit of

3600 seconds. All other solver parameters are unchanged.

We exclude instances inapplicable to `PMVB` when any of the following conditions is met:

1. $\mathcal{P}_{\mathcal{C}_\mathcal{U}, \mathcal{C}_\mathcal{L}}$ is immediately certified as infeasible;

2. both $\mathcal{P}_{\mathcal{C}_\mathcal{U}, \mathcal{C}_\mathcal{L}}$ and the original problem reach time limit.

We evaluate two datasets: combinatorial problem instances (`SCP-V` and `CA-V`) from (Gasse et al., 2019), and MIPLIB instances from (Gleixner et al., 2021). The results appear in Figure **3**, Table **2**, and Table **3**.

**Combinatorial Problems.** Figure **3** presents the speedup results for both `Gurobi` and `COPT` on two types of combinatorial problems: set covering (`SCP-V`) and combinatorial auction (`CA-V`). As shown in the figure, the data-free version of `PMVB` still achieves non-trivial speedups, demonstrating its effectiveness even without training data.

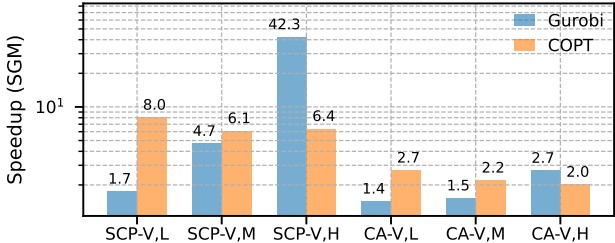

*Figure 3.* Data-free experiments on `SCP-V` and `CA-V`. The meaning of each row is the same as the second row of Figure **2**.

**MIPLIB Instances.** We ended up with 57 instances for evaluation.

*Table 2.* SGM for 57 effective MIPLIB instances

| Parameter | $\Sigma$ |
|-----------|----------|
| $\delta = 0.01, \tau = 0.9$ | $< 1.0$ (slow down) |
| $\delta = 10^{-8}, \tau = 0.9$ | 1.19 |
| $\delta = 10^{-15}, \tau = 0.95$ | 1.06 |

As shown in Table **2** and Table **3** in the appendix, we observe an improvement of up to 16% in SGM on these instances. We make the following observations:

- Some instances (e.g., `academictables` in Table **3**) become solvable on the primal side when solving $\mathcal{P}_{\mathcal{C}_\mathcal{U}, \mathcal{C}_\mathcal{L}}$. Moreover, when problem is feasible, it rarely compromises the optimality of the original problem.

- It is possible for the subproblem $\mathcal{P}_{\mathcal{C}_\mathcal{U}, \mathcal{C}_\mathcal{L}}$ to be infeasible when using the LP solution, especially when the LP relaxation is weak. In such cases, the LP solution often leads to misleading branching hyperplanes.

*Table 3.* Statistics of COPT on MIPLIB instances $\delta = 10^{-8}, \tau = 0.9$, $c^\star$ denotes the optimal objective value of solving the original problem (or $\mathcal{P}_{\mathcal{C}_\mathcal{U}, \mathcal{C}_\mathcal{L}}$), $T$ denotes solution time

| Instance | $c^\star_{\text{PMVB}}$ | $T_{\text{PMVB}}$ | $c^\star_{\text{Orig}}$ | $T_{\text{Orig}}$ | Instance | $c^\star_{\text{PMVB}}$ | $T_{\text{PMVB}}$ | $c^\star_{\text{Orig}}$ | $T_{\text{Orig}}$ |
|---|---|---|---|---|---|---|---|---|---|
| academictables | 0.0e+00 | 846.9 | 1.0e+00 | 3600.0 | ci-s4 | 3.3e+03 | 121.4 | 3.3e+03 | 101.0 |
| acc-tight5 | 0.0e+00 | 2.1 | 0.0e+00 | 4.2 | blp-ic98 | 4.5e+03 | 156.2 | 4.5e+03 | 332.1 |
| 22433 | 2.1e+04 | 0.5 | 2.1e+04 | 0.7 | brazil3 | 2.4e+02 | 70.7 | 2.4e+02 | 87.7 |
| 23588 | 8.1e+03 | 1.1 | 8.1e+03 | 0.9 | comp08-2idx | 3.7e+01 | 18.6 | 3.7e+01 | 16.2 |
| 30_70_45_05_100 | 9.0e+00 | 12.8 | 9.0e+00 | 13.6 | cmflsp50-24-8-8 | 5.6e+07 | 2361.9 | 5.6e+07 | 2764.3 |
| amaze22012-06-28i | 0.0e+00 | 0.6 | 0.0e+00 | 0.3 | diameterc-msts | 7.3e+01 | 32.4 | 7.3e+01 | 30.7 |
| arki001 | 7.5e+06 | 71.9 | 7.5e+06 | 84.2 | comp07-2idx | 6.0e+00 | 321.2 | 6.0e+00 | 468.8 |
| assign1-5-8 | 2.1e+02 | 1317.3 | 2.1e+02 | 1306.9 | aflow40b | 1.2e+03 | 88.9 | 1.2e+03 | 121.0 |
| a1c1s1 | 1.2e+04 | 204.1 | 1.2e+04 | 217.7 | cap6000 | -2.5e+06 | 0.6 | -2.5e+06 | 0.3 |
| blp-ar98 | 6.2e+03 | 293.6 | 6.2e+03 | 340.7 | ab71-20-100 | -1.0e+10 | 3.3 | -1.0e+10 | 3.5 |
| ab51-40-100 | -1.0e+10 | 6.1 | -1.0e+10 | 6.5 | beavma | 3.8e+05 | 0.2 | 3.8e+05 | 0.2 |
| a2c1s1 | 1.1e+04 | 333.2 | 1.1e+04 | 352.5 | cost266-UUE | 2.5e+07 | 1549.1 | 2.5e+07 | 1574.1 |
| blp-ar98 | 6.2e+03 | 293.6 | 6.2e+03 | 340.7 | csched007 | 3.5e+02 | 383.2 | 3.5e+02 | 408.5 |
| blp-ic97 | 4.0e+03 | 441.4 | 4.0e+03 | 465.4 | csched008 | 1.7e+02 | 80.2 | 1.7e+02 | 255.3 |
| aflow30a | 1.2e+03 | 4.4 | 1.2e+03 | 4.5 | danoint | 6.6e+01 | 650.5 | 6.6e+01 | 484.3 |

- Very conservative PMVB parameter settings are required. As suggested by Table **2**, achieving competitive performance in the data-free case requires setting the error parameter $\delta$ very close to 0. This contrasts with the data-driven case, where values of $\delta$ up to 0.3 can be used without triggering infeasibility.

## 6. Conclusions

This paper proposes PMVB, a simple yet effective method for accelerating MIP solving via machine learning. Grounded in learning theory, our approach draws a connection between concentration inequalities and multi-variable branching hyperplanes, which informs the design of a data-driven branching procedure. Even as a data-independent heuristic, PMVB also shows promising potential for general MIP solving. PMVB offers an interpretable mechanism for incorporating prior knowledge from offline solves into online MIP solving.

## Acknowledgements and Disclosure of Funding

This research is partially supported by the National Natural Science Foundation of China (NSFC) [Grant NSFC-72225009, 72394360, 72394364, 72394365, 12301403, 72192830, 72192832].

## Impact Statement

This paper presents work whose goal is to advance the field of machine learning for combinatorial optimization. There are many potential societal consequences of our work, none of which we feel must be specifically highlighted here.

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

# A. Additional Related Works

*Table 4.* Summary of `ML+MIP` approaches. GNN: graph neural networks; RL: reinforcement learning; DNN: Deep Neural Network; DQN: Deep Q-Network

| Ref | Model | Application |
|---|---|---|
| (Nair et al., 2020) | GNN | Learn to dive and branch |
| (Xavier et al., 2021) | $k$-NN and SVM | Learn affine subspaces |
| (Nazari et al., 2018) | RL | Generate near-optimal solutions |
| (Khalil et al., 2022) | GNN | Learn variable biases |
| (Gasse et al., 2019) | GNN | Learn a variable selection policy |
| (Ding et al., 2020) | GNN | Learn to predict solutions |
| (Alvarez et al., 2017) | Extremely Randomized Trees | Learn to branch |
| (Han et al., 2023) | GNN | Predict the variable probability |
| (Zarpellon et al., 2021) | GNN | Learn to generalize branching |
| (Liu et al., 2022) | GNN | Learn neighborhood size for local branching |
| (Gupta et al., 2020) | GNN, DNN, RL | Develop a GNN-MIP hybrid model |
| (Sonnerat et al., 2021) | GNN, RL, DQN, DNN | Learn to select neighborhoods for LNS |
| (Qu et al., 2022) | RL, GNN | Reinforcement learning-based branching |
| (Lin et al., 2022) | Transformer | Tree-aware transformer-based branching |

# B. Statistical Learning Theory for Data-driven MIPs

This section serves as a formal presentation of the argument from Section **2.2**. We set up the necessary assumptions that enable a more rigorous derivation of the `PMVB` framework through the lens of learning theory. We start by explaining how learning $\mathbf{y}^\star(\xi)$ fits naturally into statistical learning theory (Vapnik, 2013). The following assumptions introduce randomness to the MIP prediction task.

**A3:** (i.i.d. distribution) The instances are i.i.d. generated according to parameters drawn from a data distribution $\xi \sim \Xi$.

**A4:** (Binomial posterior estimate) Given $\xi \sim \Xi$, the MIP $\mathcal{P}(\xi)$ has one unique solution, i.e., each binary decision variable is deterministically either 0 or 1 .

Under assumption **A4**, solving an MIP can be viewed as a binary classification task. For each $j$, a binary classifier $\hat{y}_j$ takes data parameter $\xi$ as input and produces a binary prediction. The optimal classifier in this context is the Bayes classifier defined by the conditional probability $\mathbb{P}\{y_j^\star(\xi) = 1|\xi\}$. Consequently, a standard MIP solver can be viewed as one particular realization of such a Bayes classifier, which takes exponential time in the worst case.

However, classification by a MIP solver is computationally prohibitive. Instead, we select a suitable hypothesis class $\mathcal{Y}_j$ of candidate classifiers and apply empirical risk minimization (ERM) to learn an effective classifier $\hat{y}_j \in \mathcal{Y}_j$ (for example, $\hat{y}_j = \lfloor \frac{1}{2} + \tilde{y}_j \rfloor$ is a valid classifier). The learning procedure proceeds as follows: we collect offline data-solution pairs $\{(\xi_i, \mathbf{y}^\star(\xi_i))\}_{i=1}^m$, choose suitable function classes of classifiers $\mathcal{Y}_j$, and compute the empirical risk minimizer $\hat{y}_j$ via:

$$\hat{y}_j := \underset{y_j \in \mathcal{Y}_j}{\arg\min} \left\{ \tfrac{1}{m} \sum_{i=1}^m \mathbb{I}\{y_j(\xi_i) \neq y_j^\star(\xi_i)\} \right\},$$

$$e_j := \underset{y_j \in \mathcal{Y}_j}{\min} \left\{ \tfrac{1}{m} \sum_{i=1}^m \mathbb{I}\{y_j(\xi_i) \neq y_j^\star(\xi_i)\} \right\},$$

where $\mathbb{I}\{\cdot\}$ is the 0-1 indicator function, $m$ is the number of solved instances, and $e_j$ denotes ERM error. Given a new instance $\mathcal{P}(\xi)$, the trained classifier $\hat{y}_j(\xi)$ is used to predict the optimal solution component $y_j^\star(\xi)$. In practice, a convex surrogate loss function (Reid & Williamson, 2009) is employed instead of the indicator function to ensure the computational tractability of the ERM.

The performance guarantee for the classifier $\hat{y}_j$ can be characterized using standard results from statistical learning theory. Specifically, we have the following lemma:

**Lemma B.1** ((Vapnik, 2013))**.** *Under* **A3** *and* **A4***, letting $\hat{y}_j$ be the ERM classifier under 0-1 loss and $e_j$ be the ERM error, then with probability at least $1 - \delta$,*

$$\mathbb{P}_{\xi \sim \Xi}\{\hat{y}_j(\xi) \neq y_j^\star(\xi)\} \leq e_j + \sqrt{\tfrac{1}{m}\big[\,\mathrm{vc}(\mathcal{Y}_j)(\log \tfrac{2m}{\mathrm{vc}(\mathcal{Y}_j)} + 1) + \log \tfrac{4}{\delta}\big]}, \tag{14}$$

*where* $\mathrm{vc}(\mathcal{Y}_j)$ *is the VC dimension of* $\mathcal{Y}_j$.

Lemma **B.1** provides an upper bound on the prediction error of the classifier $\hat{y}_j$, showing that the upper bound improves as the sample size $m$ increases. To facilitate our analysis, we introduce an additional assumption ensuring that the trained classifier $\hat{y}_j$ is meaningful:

**A5:** (Learnability) The number of solved instances $m$ is large enough, such that for some properly chosen $\delta$ and for each $j \in [n]$, we have

$$\Delta_j^\delta := 1 - e_j - \sqrt{\tfrac{1}{m}\big[\mathrm{vc}(\mathcal{Y}_j)(\log \tfrac{2m}{\mathrm{vc}(\mathcal{Y}_j)} + 1) + \log \tfrac{4}{\delta}\big]} > 0.$$

After establishing a probabilistic interpretation of $\hat{y}_j(\xi)$ and characterizing its performance via Lemma **B.1**, we are ready to leverage these properties to introduce the PMVB idea for efficiently solving parametric MIPs.

### B.1. Risk-pooling and Multi-variable Branching

Following the previous discussions, suppose we have obtained predictions $\hat{\mathbf{y}} = (\hat{y}_1, \ldots, \hat{y}_n)$. Similar to Section **2.2**, a straightforward approach is to directly fix some binary variables according to these predictions. Specifically, for any instance $\mathcal{P}(\xi)$, we can set $y_j = \hat{y}_j(\xi)$ for each $j$ in a chosen subset $\mathcal{J} \subseteq [n]$. While this approach is straightforward, it quickly becomes unsafe – a prediction shows that the probability of correctly fixing all variables decreases exponentially when more variables are involved:

$$\mathbb{P}\{\hat{y}_j(\xi) = y_j^\star(\xi), j \in \mathcal{J}\} \geq (1 - \delta)^{|\mathcal{J}|} \textstyle\prod_{j \in \mathcal{J}} \Delta_j^\delta,$$

where $\delta$ is defined in **A5**. This observation highlights the inherent unreliability of the variable fixing strategies, where the risk of misclassification accumulates multiplicatively.

To mitigate this issue, another approach is to apply *risk pooling*, grouping these "risky" binary predictions to reduce the overall error rate. Specifically, we define the set $\mathcal{U} := \{j : \hat{y}_j(\xi) = 1\}$ and $\mathcal{L} := \{j : \hat{y}_j(\xi) = 0\}$ based on the predicted values. Using these sets, we construct two branching hyperplanes:

$$\mathcal{C}_\mathcal{U} : \textstyle\sum_{j \in \mathcal{U}} y_j \geq \zeta_1 \quad \text{and} \quad \mathcal{C}_\mathcal{L} : \textstyle\sum_{j \in \mathcal{L}} y_j \leq \zeta_2, \tag{15}$$

where $\zeta_1$ and $\zeta_2$ are parameters that need to be determined. Unlike traditional multi-variable branching strategies in the MIP literature, the probabilistic interpretation of $\{\hat{y}_j(\xi)\}$ provides a hint for selecting $\zeta_1, \zeta_2$ while characterizing the probabilistic behavior of the resulting four regions. To proceed with the analysis, we introduce the following assumption:

**A6:** (Independent error rates) For any fixed $\xi$, the events $\{\hat{y}_j(\xi) = y_j^\star(\xi)\}$ for different indices $j \in [n]$ are independent.

*Remark* B.2. Assumption **A6**, while strong, is a standard assumption in the literature on ensemble methods in machine learning (Freund & Schapire, 1997). Here, we utilize it to highlight the effectiveness of the risk pooling concept. In Section **3**, we will demonstrate that this assumption can be removed without compromising the effectiveness of our approach.

The probabilistic interpretation, combined with the independence assumption **A6**, enables us to interpret the two hyperplanes in (15) as concentration inequalities:

**Theorem B.3.** *Under the same assumptions as Lemma B.1, along with* **A5** *and* **A6***, consider a new MIP instance* $\mathcal{P}(\xi)$ *with* $\xi \sim \Xi$*. Letting* $\mathcal{U} = \{j : \hat{y}_j(\xi) = 1\}$ *and* $\mathcal{L} = \{j : \hat{y}_j(\xi) = 0\}$*, for any* $\gamma \geq 0$*, we have*

$$\mathbb{P}\big\{\textstyle\sum_{j \in \mathcal{U}} y_j^\star(\xi) \geq (1 - \delta)\textstyle\sum_{j \in \mathcal{U}} \Delta_j^\delta - \gamma\big\} \geq 1 - \exp\big(-\tfrac{2\gamma^2}{|\mathcal{U}|}\big)$$

$$\mathbb{P}\big\{\textstyle\sum_{j \in \mathcal{L}} y_j^\star(\xi) \leq |\mathcal{L}| - (1 - \delta)\textstyle\sum_{j \in \mathcal{L}} \Delta_j^\delta + \gamma\big\} \geq 1 - \exp\big(-\tfrac{2\gamma^2}{|\mathcal{L}|}\big)$$

*where the probability is taken over both the training set and the new instance.*

## C. Auxiliary Lemmas

**Lemma C.1** (Hoeffding inequality)**.** *Given independent random variables* $Y_1, \ldots, Y_n$ *such that* $Y_i \in [0, 1]$*, then for any* $t > 0$*, we have*

$$\mathbb{P}\{\textstyle\sum_{i=1}^n (Y_i - \mathbb{E}[Y_i]) \geq t\} \leq e^{-\frac{2t^2}{n}}.$$

**Lemma C.2** (Bernstein inequality). *Given i.i.d. Bernoulli random variables $Y_1, \ldots, Y_n$ with $Y_i \in \{0, 1\}$ and $\mathbb{E}[Y_i] = p$, then for any $t > 0$, we have*

$$\mathbb{P}\{\textstyle\sum_{i=1}^n Y_i - np \geq t\} \leq \exp(-\tfrac{t^2}{2(np+t/3)}).$$

**Lemma C.3** (Chebyshev inequality). *Given a random variable $Y$ such that $\mathbb{V}[Y] < \infty$, then for any $t > 0$, we have*

$$\mathbb{P}\{|Y - \mathbb{E}[Y]| \geq t\} \leq \tfrac{\mathbb{V}[Y]}{t^2}.$$

**Lemma C.4.** *Given a sequence $\{\lambda_i, i = 1, 2, \ldots, n\}$ which are independent and uniformly distributed in $(0, 1)$, for any fixed $\delta \in (0, 1)$ and any $1 \leq j \leq 1/\delta$, define the set $I_j = \{i : \lambda_i \in [(j-1)\delta, j\delta]\}$, we have that with probability at least $1 - \lceil \frac{1}{\delta} \rceil \exp(\frac{-n\delta}{4})$, we have*

$$|I_j| \leq 2n\delta \text{ for all } 1 \leq j \leq \lceil 1/\delta \rceil.$$

*Besides, for all subsets $S \subseteq [n]$ satisfying $|S| \geq 4n\delta$, we have $\sum_{i \in S} \lambda_i \geq \frac{|S|^2}{8n}$.*

*Proof.* For any fixed $j$, we know that $\mathbb{P}(\lambda_i \in [(j-1)\delta, j\delta]) = \delta$ for all $i \in [n]$. Then, by applying Lemma **C.2** with $p = \delta$ and $t = n\delta$, we could show that

$$\mathbb{P}\{|\{i : \lambda_i \in [(j-1)\delta, j\delta]\}| \geq 2n\delta\} \leq \exp(\tfrac{-n\delta}{4}). \tag{16}$$

According to the union bound, (16) holds for all $1 \leq i \leq 1/\delta$ with probability at least $1 - \frac{1}{\delta} \exp(\frac{-n\delta}{4})$. Besides, consider the minimum of the sum $\sum_{i \in S} \lambda_i$ among all subsets $S \subseteq [n]$, which is achieved as we choose $S = I_1 \cup I_2 \cup \cdots \cup I_k$ for some $k$ and each $|I_j|$ is as large as possible, i.e., $|I_j| = 2n\delta$ and $k = |S|/(2n\delta)$. So we have

$$\textstyle\sum_{i \in S} \lambda_i \geq \sum_{j=1}^{|S|/(2n\delta)} 2(j-1)n\delta^2 \geq \tfrac{|S|}{4n}(|S| - 2n\delta) \geq \tfrac{|S|^2}{8n}$$

where the last inequality holds because we assume $|S| \geq 4n\delta$. $\qquad\square$

## D. Missing Proofs

### D.1. Proof of Lemma B.1

For any $\xi \in \Xi$ and any $j \in \mathcal{U}$, we have

$$\mathbb{E}[y_j^\star(\xi)] = \mathbb{P}\{y_j^\star(\xi) = 1\} = \mathbb{P}\{y_j^\star(\xi) = \hat{y}_j(\xi)\} \geq (1 - \delta)\Delta_j^\delta, \tag{17}$$

where the first equality holds because $y_j^\star(\xi)$ is a binary variable, the second equality holds because $\hat{y}_j(\xi) = 1$ for any $j \in \mathcal{U}$, and the last inequality holds since Lemma **B.1** suggests that (14) holds with probability at least $1 - \delta$. Then, based on **A6**, we could apply the Hoeffding's inequality (see Lemma **C.1** in Appendix) to the sum $\sum_{j \in \mathcal{U}} y_j^\star(\xi)$ to get

$$\mathbb{P}\big\{\textstyle\sum_{j \in \mathcal{U}} y_j^\star(\xi) \geq \sum_{j \in \mathcal{U}} \mathbb{E}[y_j^\star(\xi)] - \gamma\big\} \geq 1 - \exp\big(-\tfrac{2\gamma^2}{|\mathcal{U}|}\big). \tag{18}$$

We complete the proof of the first part by combining (17) and (18) together. The proof of the second part is almost the same, so we omit it here.

### D.2. Proof of Theorem B.3

The side $j \in \mathcal{L}$ follows by

$$\begin{aligned}
\mathbb{P}\{\textstyle\sum_{j \in \mathcal{L}} y_j^\star(\xi) \leq \eta\} &= \mathbb{P}\{\textstyle\sum_{j \in \mathcal{L}} y_j^\star(\xi) - \hat{y}_j(\xi) \leq \eta\} \\
&= \mathbb{P}\{\textstyle\sum_{j \in \mathcal{L}} 1 - \mathbb{I}\{y_j^\star(\xi) = \hat{y}_j(\xi)\} \leq \eta\} \\
&= \mathbb{P}\{\textstyle\sum_{j \in \mathcal{L}} \mathbb{I}\{y_j^\star(\xi) = \hat{y}_j(\xi)\} \geq |\mathcal{L}| - \eta\} \\
&= \mathbb{P}\{\textstyle\sum_{j \in \mathcal{L}} \mathbb{I}\{y_j^\star(\xi) = \hat{y}_j(\xi)\} - (1-\delta)\sum_{j \in \mathcal{L}} \Delta_j^\delta \geq |\mathcal{L}| - \eta - (1-\delta)\sum_{j \in \mathcal{L}} \Delta_j^\delta\} \\
&\geq \mathbb{P}\{\textstyle\sum_{j \in \mathcal{L}} \mathbb{I}\{y_j^\star(\xi) = \hat{y}_j(\xi)\} - \pi_j \geq |\mathcal{L}| - \eta - (1-\delta)\sum_{j \in \mathcal{L}} \Delta_j^\delta\} \\
&\geq 1 - \mathbb{P}\{\textstyle\sum_{j \in \mathcal{L}} \mathbb{I}\{y_j^\star(\xi) = \hat{y}_j(\xi)\} - \pi_j \leq |\mathcal{L}| - \eta - (1-\delta)\sum_{j \in \mathcal{L}} \Delta_j^\delta\} \\
&\geq 1 - \exp(-2\tfrac{[|\mathcal{L}| - \eta - (1-\delta)\sum_{j \in \mathcal{L}} \Delta_j^\delta]}{|\mathcal{L}|}).
\end{aligned}$$

Taking $\eta = |\mathcal{L}| - (1-\delta)\sum_{j \in \mathcal{L}} \Delta_j^\delta + \gamma$ completes the proof.

### D.3. Proof of Theorem 4.1

The LP relaxation of (13) is given by

$$\max_{\mathbf{y} \in \mathbb{R}^n} \quad c_1 y_1 + \cdots + c_n y_n$$
$$\text{subject to} \quad a_1 y_1 + \cdots + a_n y_n \le b, \tag{19}$$
$$0 \le y_i \le 1, \quad \text{for all } i \in [n].$$

We could write down its Lagrangian function as

$$L(\mathbf{y}, \lambda, \eta, \xi) = -(c_1 y_1 + \cdots + c_n y_n) + \lambda(a_1 y_1 + \cdots + a_n y_n - b) + \sum_{i=1}^{n} \eta_i(y_i - 1) - \sum_{i=1}^{n} \xi_i y_i$$

with $\lambda \ge 0$, $\eta \ge 0$ and $\xi \ge 0$. Denote by $\tilde{\mathbf{y}}$ and $(\lambda^\star, \eta^\star, \xi^\star)$ the primal and dual optimal solutions to (19), which should satisfy the following KKT optimality conditions:

$$a_i(f_i - \lambda^\star) = \eta_i^\star - \xi_i^\star, \quad \text{for all } i$$
$$\eta_i^\star(y_i - 1) = 0, \quad \text{for all } i$$
$$\xi_i^\star y_i = 0, \quad \text{for all } i.$$

Then, it is easy to see that the optimal solution $\tilde{\mathbf{y}}$ is given by

$$\tilde{y}_i \in \begin{cases} \{1\} & \text{if } f_i > \lambda^\star, \\ \{0\} & \text{if } f_i < \lambda^\star, \\ [0, 1] & \text{otherwise.} \end{cases} \tag{20}$$

Without loss of generality, we assume $f_1 > f_2 > \cdots > f_n$. Recall the definition $\mathcal{U} := \{i : \tilde{y}_i = 1\}$ and $\mathcal{L} = \{i : \tilde{y}_i = 0\}$. Then, (20) implies that there exists some constant $k \in [n]$ such that

$$\mathcal{U} = \{1, 2, \ldots, k\} \quad \text{and} \quad \mathcal{L} = \{k+2, \ldots, n\}.$$

Here, we assume $\tilde{y}_{k+1} \in (0, 1)$. Otherwise, $\tilde{\mathbf{y}}$ is already the optimal solution to the original knapsack problem (13), and the result trivially holds. Let $\mathcal{U}^\star := \{i : y_i^\star = 1\}$ and $\mathcal{L}^\star := \{i : y_i^\star = 0\}$. According to the optimality condition of $\mathbf{y}^\star$, we have

$$c_1 y_1^\star + \cdots + c_n y_n^\star = \sum_{i \in \mathcal{U}^\star} c_i \ge \sum_{i \in \mathcal{U}} c_i$$
$$a_1 y_1^\star + \cdots + a_n y_n^\star = \sum_{i \in \mathcal{U}^\star} a_i \le b \le \sum_{i \in \mathcal{U}} a_i + 1,$$

where the last inequality holds since $b = a_1 \tilde{y}_1 + \cdots + a_n \tilde{y}_n = \sum_{i \in \mathcal{U}} a_i + a_{k+1} \tilde{y}_{k+1} \le \sum_{i \in \mathcal{U}} a_i + 1$. In other words,

$$\sum_{i \in \mathcal{U}^\star \setminus \mathcal{U}} c_i \ge \sum_{i \in \mathcal{U} \setminus \mathcal{U}^\star} c_i \quad \text{and} \quad \sum_{i \in \mathcal{U}^\star \setminus \mathcal{U}} a_i \le 1 + \sum_{i \in \mathcal{U} \setminus \mathcal{U}^\star} a_i \tag{21}$$

Since $\mathcal{U} = \{1, 2, \ldots, k\}$, we know that $\mathcal{U}^\star \setminus \mathcal{U} \subseteq \{k+1, \cdots, n\}$. A simple computation shows

$$\sum_{i \in \mathcal{U}^\star \setminus \mathcal{U}} c_i = \sum_{i \in \mathcal{U}^\star \setminus \mathcal{U}} f_i \cdot a_i \le (f_{k+1} - \delta) \sum_{i \in \mathcal{U}^\star \setminus \mathcal{U}} a_i + \delta \cdot |\{i : f_i \in [f_{k+1} - \delta, f_{k+1}]\}|. \tag{22}$$

Note that for any given $\delta$, we know that $[f_{k+1} - \delta, f_{k+1}] \subset [(j-1)\delta, (j+1)\delta]$ for some $1 \le j \le 1/\delta$. Then, by applying Lemma **C.4** with $2\delta$, we know that

$$|\{i : f_i \in [f_{k+1} - \delta, f_{k+1}]\}| \le 4n\delta \tag{23}$$

holds with probability at least $1 - \lceil \frac{1}{\delta} \rceil \exp(\frac{-n\delta}{2})$. Combining (21), (22) and (23) together, we get

$$\sum_{i \in \mathcal{U}^\star \setminus \mathcal{U}} c_i \le (f_{k+1} - \delta) \sum_{i \in \mathcal{U}^\star \setminus \mathcal{U}} a_i + 4\delta^2 n \le (f_{k+1} - \delta)(1 + \sum_{i \in \mathcal{U} \setminus \mathcal{U}^\star} a_i) + 4\delta^2 n. \tag{24}$$

Next, the fact that $\sum_{i \in \mathcal{U}^\star \setminus \mathcal{U}} c_i \ge \sum_{i \in \mathcal{U} \setminus \mathcal{U}^\star} c_i \ge f_k \cdot \sum_{i \in \mathcal{U} \setminus \mathcal{U}^\star} a_i$ implies

$$f_k \cdot \sum_{i \in \mathcal{U} \setminus \mathcal{U}^\star} a_i \le (f_{k+1} - \delta)(1 + \sum_{i \in \mathcal{U} \setminus \mathcal{U}^\star} a_i) + 4\delta^2 n \le 1 + (f_k - \delta) \sum_{i \in \mathcal{U} \setminus \mathcal{U}^\star} a_i + 4\delta^2 n.$$

A simple computation shows that

$$\sum_{i \in \mathcal{U} \setminus \mathcal{U}^\star} a_i \le \frac{1}{\delta} + 4\delta n = 4\sqrt{n}$$

where the last inequality holds by choosing $\delta = \frac{1}{2\sqrt{n}}$. Lastly, by applying Lemma **C.4** with $\delta = \frac{1}{2\sqrt{n}}$ and $S = \mathcal{U} \setminus \mathcal{U}^\star$, we get $|\mathcal{U} \setminus \mathcal{U}^\star| \leq 4\sqrt{2}n^{\frac{3}{4}}$. Thus, we complete the proof by

$$\sum_{i \in \mathcal{U}} y_i^\star \geq \sum_{i \in \mathcal{U} \cap \mathcal{U}^\star} y_i^\star = \sum_{i \in \mathcal{U} \cap \mathcal{U}^\star} \tilde{y}_i = \sum_{i \in \mathcal{U}} \tilde{y}_i - |\mathcal{U} \setminus \mathcal{U}^\star| \geq \sum_{i \in \mathcal{U}} \tilde{y}_i - 4\sqrt{2}n^{\frac{3}{4}}$$

where the first equality holds since $y_i^\star = \tilde{y}_i = 1$ for all $i \in \mathcal{U} \cap \mathcal{U}^\star$. For the inequality with respect to $\mathcal{L}$, the same constant does not follow directly from complementarity. By (21) and the bound above,

$$\sum_{i \in \mathcal{U}^\star \setminus \mathcal{U}} a_i \leq 1 + \sum_{i \in \mathcal{U} \setminus \mathcal{U}^\star} a_i \leq 1 + 4\sqrt{n}.$$

Let $S = \mathcal{U}^\star \setminus \mathcal{U}$. If $|S| \geq 4\sqrt{2}n^{\frac{3}{4}} + n^{\frac{1}{4}}$, then by Lemma **C.4** with $\delta = \frac{1}{2\sqrt{n}}$,

$$\sum_{i \in \mathcal{U}^\star \setminus \mathcal{U}} a_i \geq \frac{|S|^2}{8n} \geq \frac{(4\sqrt{2}n^{\frac{3}{4}} + n^{\frac{1}{4}})^2}{8n} \geq 4\sqrt{n} + \sqrt{2},$$

which contradicts the previous upper bound. Hence $|\mathcal{U}^\star \setminus \mathcal{U}| \leq 4\sqrt{2}n^{\frac{3}{4}} + n^{\frac{1}{4}}$. Since $\mathcal{L} \cap \mathcal{U}^\star \subseteq ([n] \setminus \mathcal{U}) \cap \mathcal{U}^\star = \mathcal{U}^\star \setminus \mathcal{U}$, we have

$$\sum_{i \in \mathcal{L}} y_i^\star = |\mathcal{L} \cap \mathcal{U}^\star| \leq |\mathcal{U}^\star \setminus \mathcal{U}| \leq 4\sqrt{2}n^{\frac{3}{4}} + n^{\frac{1}{4}}.$$

### D.4. Data-free `PMVB` for Multi-knapsack

We refine the data-free PMVB argument for the multi-knapsack problem

$$z_{\mathrm{IP}} := \max_{\mathbf{y} \in \{0,1\}^n} \langle \mathbf{c}, \mathbf{y} \rangle \quad \text{subject to} \quad \mathbf{A}\mathbf{y} \leq \mathbf{b}, \quad \text{where } \mathbf{A} = [\mathbf{a}_1, \dots, \mathbf{a}_n] \in \mathbb{R}_+^{m \times n}, \quad \mathbf{b} \in \mathbb{R}_+^m. \tag{25}$$

Let $z_{\mathrm{LP}}$ be the optimal value of the LP relaxation

$$\max_{\mathbf{0} \leq \mathbf{y} \leq \mathbf{1}} \langle \mathbf{c}, \mathbf{y} \rangle : \mathbf{A}\mathbf{y} \leq \mathbf{b}.$$

Let $\tilde{\mathbf{y}}$ be an optimal basic feasible solution of the LP relaxation, and define

$$U := \{i : \tilde{y}_i = 1\}, \qquad L := \{i : \tilde{y}_i = 0\}, \qquad F := \{i : 0 < \tilde{y}_i < 1\}.$$

Since the relaxation has $m$ knapsack constraints, every basic feasible solution has at most $m$ non-bound variables. Hence

$$|F| \leq m. \tag{26}$$

**Assumption D.1** (Smoothed data model). Let $X_i = (c_i, \mathbf{a}_i)$, $i = 1, \dots, n$, denote the item data. Assume:

(i) $X_1, \dots, X_n$ are independent.

(ii) There exist constants $\bar{a}, C_{\max}, \beta > 0$, independent of $n$, such that

$$0 \leq a_{ji} \leq \bar{a}, \qquad 0 \leq c_i \leq C_{\max}, \qquad b_j \geq \beta n, \quad j = 1, \dots, m.$$

(iii) Conditional on $\mathbf{a}_i$, the profit $c_i$ has a density $f_{c_i | \mathbf{a}_i}$ satisfying

$$\sup_{i,t,\mathbf{a}} f_{c_i | \mathbf{a}_i = \mathbf{a}}(t) \leq \rho.$$

Define

$$\Lambda_0 := \frac{C_{\max}}{\beta}, \qquad \Lambda := [0, \Lambda_0]^m, \qquad A_1 := m\bar{a}.$$

**Lemma D.2** (Uniform reduced-cost boundary bound). *For $\lambda \in \Lambda$, define*

$$B_\varepsilon(\lambda) := \{i \in [n] : |c_i - \langle \mathbf{a}_i, \lambda \rangle| \leq \varepsilon\}.$$

*Under Assumption **D.1**, every optimal dual multiplier $\lambda^\star$ of the LP relaxation can be chosen from $\Lambda$. Moreover, for any $\varepsilon > 0$ and $\eta \in (0,1)$, with probability at least $1 - \eta$,*

$$\sup_{\lambda \in \Lambda} |B_\varepsilon(\lambda)| \leq \mathcal{B}(\varepsilon, \eta), \tag{27}$$

*where*

$$\mathcal{B}(\varepsilon, \eta) := 4\rho\varepsilon n + \sqrt{\frac{n}{2}[m \log(1 + \frac{2\Lambda_0 A_1}{\varepsilon}) + \log \frac{1}{\eta}]}. \tag{28}$$

*Proof.* The dual of the LP relaxation can be written as

$$z_{\mathrm{LP}} = \min_{\lambda \geq 0} g(\lambda) := \langle \mathbf{b}, \lambda \rangle + \sum_{i=1}^{n}(c_i - \langle \lambda, \mathbf{a}_i \rangle)_+,$$

where $(\cdot)_+$ denotes the positive part function. Since $g(0) = \sum_i c_i \leq nC_{\max}$, any optimal dual multiplier $\lambda^\star$ satisfies

$$\langle \mathbf{b}, \lambda^\star \rangle \leq g(\lambda^\star) \leq g(0) \leq nC_{\max}.$$

Using $b_j \geq \beta n$, we obtain

$$\lambda_j^\star \leq \frac{nC_{\max}}{b_j} \leq \frac{C_{\max}}{\beta} = \Lambda_0, \qquad j = 1, \ldots, m.$$

Thus $\lambda^\star \in \Lambda$, and it remains to prove the uniform boundary bound. Let $\mathcal{N}_h$ be an $h$-net of $\Lambda$ under $\| \cdot \|_\infty$, chosen so that

$$|\mathcal{N}_h| \leq (1 + \tfrac{2\Lambda_0}{h})^m.$$

For every $\lambda \in \Lambda$, choose $\hat{\lambda} \in \mathcal{N}_h$ satisfying $\|\lambda - \hat{\lambda}\|_\infty \leq h$. Since $\|\mathbf{a}_i\|_1 \leq m\bar{a} = A_1$,

$$|c_i - \langle \lambda, \mathbf{a}_i \rangle| \leq \varepsilon \quad \Longrightarrow \quad |c_i - \langle \hat{\lambda}, \mathbf{a}_i \rangle| \leq \varepsilon + A_1 h.$$

Hence

$$B_\varepsilon(\lambda) \subseteq B_{\varepsilon + A_1 h}(\hat{\lambda}), \qquad \sup_{\lambda \in \Lambda} |B_\varepsilon(\lambda)| \leq \max_{\hat{\lambda} \in \mathcal{N}_h} |B_{\varepsilon + A_1 h}(\hat{\lambda})|.$$

Fix $\hat{\lambda} \in \mathcal{N}_h$, and define

$$Z_i(\hat{\lambda}) := \mathbf{1}\{|c_i - \langle \hat{\lambda}, \mathbf{a}_i \rangle| \leq \varepsilon + A_1 h\}.$$

By the bounded-density assumption,

$$\mathbb{E}[Z_i(\hat{\lambda})] \leq 2\rho(\varepsilon + A_1 h).$$

Since the item data are independent, Hoeffding's inequality gives

$$\mathbb{P}\{\textstyle\sum_{i=1}^{n} Z_i(\hat{\lambda}) \geq 2\rho(\varepsilon + A_1 h)n + t\} \leq \exp(-\tfrac{2t^2}{n}).$$

Taking a union bound over $\mathcal{N}_h$, with probability at least $1 - \eta$,

$$\max_{\hat{\lambda} \in \mathcal{N}_h} \textstyle\sum_{i=1}^{n} Z_i(\hat{\lambda}) \leq 2\rho(\varepsilon + A_1 h)n + \sqrt{\tfrac{n}{2}[m\log(1 + \tfrac{2\Lambda_0}{h}) + \log\tfrac{1}{\eta}]}.$$

Choosing $h = \varepsilon/A_1$ completes the proof of (27). $\qquad\square$

**Theorem D.3** (Data-free `PMVB` guarantee for multi-knapsack)**.** *Let $y^\star$ be an optimal solution of (25). Let $\lambda^\star \in \Lambda$ be an optimal dual multiplier compatible with the LP optimal basic feasible solution $\tilde{y}$, and define the reduced costs*

$$\Delta_i := c_i - \langle \mathbf{a}_i, \lambda^\star \rangle.$$

*Since $0 \leq c_i \leq C_{\max}$, the LP–IP gap satisfies*

$$\Gamma := z_{\mathrm{LP}} - z_{\mathrm{IP}} \leq mC_{\max}.$$

*Under Assumption **D.1**, for any $\varepsilon > 0$ and $\eta \in (0, 1)$, with probability at least $1 - \eta$,*

$$\sum_{i \in U} y_i^\star \geq |U| - R_\Gamma(\varepsilon, \eta), \tag{29}$$

$$\sum_{i \in L} y_i^\star \leq R_\Gamma(\varepsilon, \eta), \tag{30}$$

*where*

$$R_\Gamma(\varepsilon, \eta) := \mathcal{B}(\varepsilon, \eta) + \frac{mC_{\max}}{\varepsilon}. \tag{31}$$

*Proof.* By the KKT conditions of the LP relaxation,

$$i \in U \Rightarrow \Delta_i \geq 0, \qquad i \in L \Rightarrow \Delta_i \leq 0, \qquad i \in F \Rightarrow \Delta_i = 0.$$

Define the two mistake sets

$$M_U := \{i \in U : y_i^\star = 0\}, \qquad M_L := \{i \in L : y_i^\star = 1\}.$$

By LP strong duality,

$$z_{\mathrm{LP}} = \langle \mathbf{b}, \lambda^\star \rangle + \sum_{i=1}^n (\Delta_i)_+.$$

For the integer optimal solution $\mathbf{y}^\star$,

$$z_{\mathrm{IP}} = \langle \mathbf{c}, \mathbf{y}^\star \rangle = \langle \lambda^\star, \mathbf{A}\mathbf{y}^\star \rangle + \sum_{i=1}^n \Delta_i y_i^\star.$$

Therefore,

$$\begin{aligned}
\Gamma &= z_{\mathrm{LP}} - z_{\mathrm{IP}} \\
&= \langle \lambda^\star, \mathbf{b} - \mathbf{A}\mathbf{y}^\star \rangle + \sum_{\Delta_i > 0} \Delta_i(1 - y_i^\star) + \sum_{\Delta_i < 0} (-\Delta_i) y_i^\star.
\end{aligned} \tag{32}$$

The first term is nonnegative because $\lambda^\star \geq 0$ and $Ay^\star \leq b$.

Let

$$B_\varepsilon(\lambda^\star) := \{i \in [n] : |\Delta_i| \leq \varepsilon\}.$$

If $i \in M_U \setminus B_\varepsilon(\lambda^\star)$, then $\Delta_i > \varepsilon$. If $i \in M_L \setminus B_\varepsilon(\lambda^\star)$, then $\Delta_i < -\varepsilon$. Hence (32) implies

$$\varepsilon(|M_U \setminus B_\varepsilon(\lambda^\star)| + |M_L \setminus B_\varepsilon(\lambda^\star)|) \leq \Gamma.$$

Thus

$$|M_U| \leq |B_\varepsilon(\lambda^\star)| + \tfrac{\Gamma}{\varepsilon}, \qquad |M_L| \leq |B_\varepsilon(\lambda^\star)| + \tfrac{\Gamma}{\varepsilon}.$$

Lemma **D.2** holds uniformly over all $\lambda \in \Lambda$. Since $\lambda^\star \in \Lambda$, it can be evaluated at the random, data-dependent multiplier $\lambda^\star$, yielding

$$|B_\varepsilon(\lambda^\star)| \leq \mathcal{B}(\varepsilon, \eta)$$

with probability at least $1 - \eta$. Therefore,

$$|M_U|, |M_L| \leq R_\Gamma(\varepsilon, \eta).$$

Finally,

$$\sum_{i \in U} y_i^\star = |U| - |M_U| \geq |U| - R_\Gamma(\varepsilon, \eta), \qquad \sum_{i \in L} y_i^\star = |M_L| \leq R_\Gamma(\varepsilon, \eta).$$

It remains to prove the explicit gap bound. Define

$$\bar{y}_i := \mathbf{1}\{\tilde{y}_i = 1\}.$$

Since $\mathbf{A} \geq 0$ and $\bar{\mathbf{y}} \leq \tilde{\mathbf{y}}$ componentwise, we have

$$\mathbf{A}\bar{\mathbf{y}} \leq \mathbf{A}\tilde{\mathbf{y}} \leq \mathbf{b}.$$

Hence $\bar{\mathbf{y}}$ is feasible for the integer problem. Hence

$$z_{\mathrm{IP}} \geq \langle \mathbf{c}, \bar{\mathbf{y}} \rangle = \langle \mathbf{c}, \tilde{\mathbf{y}} \rangle - \sum_{i \in F} c_i \tilde{y}_i \geq z_{\mathrm{LP}} - C_{\max}|F|.$$

$\square$

*Remark* D.4 (Rate). Choosing

$$\varepsilon = \sqrt{\frac{mC_{\max}}{4\rho n}}$$

balances the two deterministic terms $4\rho\varepsilon n$ and $mC_{\max}/\varepsilon$. Therefore, up to logarithmic factors,

$$R_\Gamma(\varepsilon, \eta) = \mathcal{O}\left(\sqrt{\rho m C_{\max} n} + \sqrt{n(m \log n + \log \tfrac{1}{\eta})}\right).$$

In particular, for fixed $m, \rho, C_{\max}, \bar{a}, \beta$, the PMVB prediction error is $\mathcal{O}(\sqrt{n \log n})$, hence sublinear in $n$.

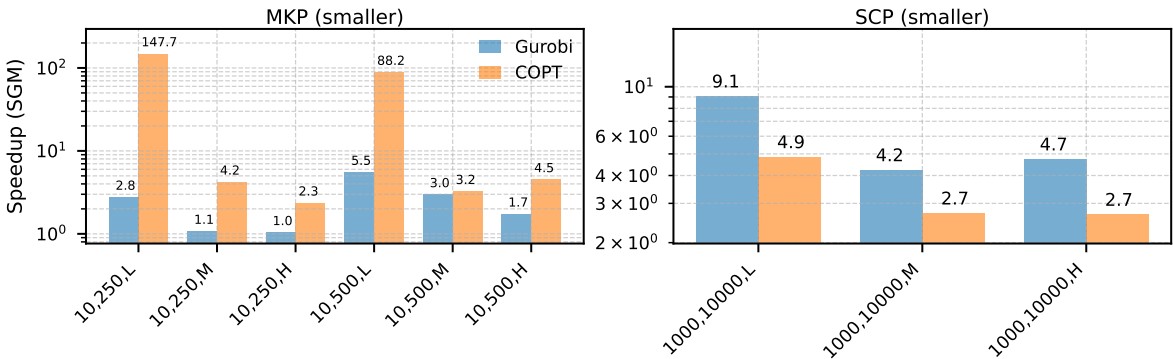

*Figure 4.* Additional logistic regression speedup results omitted from Figure **2** for compactness.

# E. Additional Experiments

### E.1. Additional Figures

### E.2. Effect of Different GNN Architectures

**Experiment Setup.** In this section, we conduct ablations on the effect of using different GNN architectures. We consider the following three GNN architectures:

- `MIPGNN` (Cantürk et al., 2024)

- Heterogeneous Graph Transformer (`HGT`) (Hu et al., 2020)

- `FMIP` (Li et al., 2025)

We report results on `SCP-V` and `IS-V` datasets with `Gurobi` and default Heuristic level (0.05). We report the testing accuracy and the speedup $\Sigma$ compared to `Gurobi` with warm start.

*Table 5.* Comparison between different GNN prediction architectures

| Architecture | SCP-V accuracy | SCP-V speedup $\Sigma$ | IS-V accuracy | IS-V speedup $\Sigma$ |
|---|---|---|---|---|
| MIPGNN | 96.98% | 2.21 | 83.53% | 51.12 |
| HGT | 97.15% | 3.44 | 86.84% | 188.20 |
| FMIP | 98.18% | 3.80 | 89.35% | 233.95 |

The experiments suggest that on both problem types, the final speedup has a positive correlation with the prediction accuracy, and using more advanced upstream model can benefit `PMVB`.

### E.3. Comparison between PMVB & P&S

**Experiment Setup.** In this section, we conduct experiments to compare the performance between `PMVB` and `P&S` (Han et al., 2023). We reuse the testing environment from (Li et al., 2025). We adopt two testing datasets from the NeurIPS (Gasse et al., 2022a) `ML4CO` competition:

- Item placement. See Section 4.1 of (Gasse et al., 2022a) for more details

- Load balancing. See Section 4.2 of (Gasse et al., 2022a) for more details

Both type of problems have 10000 instances in the dataset. There are also 100 testing instances for each. The following upstream prediction models are used, and the model weights are directly taken from (Li et al., 2025):

- Binary GCN from (Wang et al., 2021)

- GCN from (Gasse et al., 2022a)

- Graph attention network (Veličković et al., 2018)

- ClusterGCN (Chiang et al., 2019)

For each testing instance, we run `PMVB` and `P&S` for 600 seconds and report the final objective value using different models.

*Table 6.* Comparison of objective value after 600 seconds.

| Instance | Load Balancing Problem ($\downarrow$) | | | | Item Placement Problem ($\downarrow$) | | | |
|---|---|---|---|---|---|---|---|---|
| Method | Binary GCN | GCN | Graph Attention | ClusterGCN | Binary GCN | GCN | Graph Attention | ClusterGCN |
| P&S | 750.34 | 749.60 | 749.26 | 749.69 | 15.38 | 15.34 | 15.33 | 15.32 |
| PMVB | 706.50 | 706.10 | 706.30 | 706.30 | 15.36 | 15.39 | 15.36 | 15.44 |

For the load balancing problem with partitioning constraint $\sum_j y_j = 1$, `PMVB` in general outperforms `P&S`, while on item placement, the two methods are comparable.

### E.4. Comparison between PMVB and Solvers without Warmstart

This section complements our experiments by turning off the solver warm start feature. Our algorithm experiment baseline already makes use of `Gurobi`'s `Start` and `COPT`'s `Mip Start` interfaces to start from the rounded solution prediction. The experiment setting are the same as in the paper, and we report the speedup statistics $\Sigma$ compared to solvers without warm start.

*Table 7.* Speedup statistics of `PMVB` using `COPT`

| Instance | Heuristics | $\Sigma$ w.r.t. solver with warm start | $\Sigma$ w.r.t. solver without warm start |
|---|---|---|---|
| SCP-V | L | 10.8 | 10.6 |
| SCP-V | M | 8.3 | 8.3 |
| SCP-V | H | 9.2 | 7.6 |
| IS-V | L | 22.5 | 38.6 |
| IS-V | M | 42.7 | 49.7 |
| IS-V | H | 297.5 | 177.1 |
| CA-V | L | 2.7 | 3.1 |
| CA-V | M | 2.2 | 2.3 |
| CA-V | H | 2.1 | 2.2 |

*Table 8.* Speedup statistics of `PMVB` using `Gurobi`

| Instance | Heuristics | $\Sigma$ w.r.t. solver with warm start | $\Sigma$ w.r.t. solver without warm start |
|---|---|---|---|
| SCP-V | L | 2.2 | 1.8 |
| SCP-V | M | 6.8 | 2.9 |
| SCP-V | H | 47.3 | 29.0 |
| IS-V | L | 51.1 | 91.2 |
| IS-V | M | 261.4 | 554.1 |
| IS-V | H | 572.8 | 891.2 |
| CA-V | L | 1.2 | 1.2 |
| CA-V | M | 1.3 | 1.1 |
| CA-V | H | 2.7 | 1.7 |

There are observations from the experiments

- The trends of speedup with(out) warm start are consistent across solvers and datasets.

- On set covering instances, adding warm start does not seem to benefit solvers, as speedup statistics are largely unaffected or even increased (hence version with warm start is even relatively slower)

- On independent set instances, `COPT` does not benefit from warm start, while `Gurobi` does.

- On combinatorial auction instances, `COPT` benefits from warm start, while `Gurobi` doesn't.

Hence the effect of adding warm start based on rounded predictions has very different effects on different solvers. But the acceleration effect of `PMVB` persists in both cases.

### E.5. Empirical Studies on Accuracy of LP Relaxation Solution

This section investigates how well the solution to LP relaxation of a MIP can be used as a proxy of the true binary solution.

**Experiment Setup.** We choose 90 small to medium scale MIPLIB instances. For each instance, we use `Gurobi` to

- Solve the LP relaxation using simplex/interior point method and obtain $\hat{\mathbf{y}}_{\text{Simplex}}$ and $\hat{\mathbf{y}}_{\text{IPM}}$ as prediction of the true solutions.

- Solve the MIP problem to a gap of 0.1 and obtain the binary solution.

- Compute the prediction accuracy $\frac{1}{n} \sum_{j=1}^{n} \mathbb{I}\{\lfloor \hat{y}_j + \frac{1}{2} \rfloor = y_j^\star\}$.

We end up getting 77 feasible instances, and the statistics over them are given below:

*Table 9.* Accuracy of LP solution

| Metric/Algorithm | Interior point method | Simplex |
|---|---|---|
| Mean accuracy | 82.9% | 81.1% |
| Median accuracy | 88.1% | 83.9% |

*Table 10.* Distribution of accuracy

| Accuracy | Interior point method | Simplex |
|---|---|---|
| $< 50\%$ | 4 | 4 |
| $50 \sim 70\%$ | 11 | 12 |
| $70 \sim 80\%$ | 14 | 18 |
| $80 \sim 90\%$ | 13 | 9 |
| $90 \sim 95\%$ | 12 | 13 |
| $> 95\%$ | 23 | 21 |

The results indicate that

- LP solutions are relatively reliable for $> 80\%$ of the MIPLIB instances, with a prediction accuracy around $80\%$. There exist instances ($< 20\%$) where LP relaxations are non-informative. The overall prediction power is moderate-to-strong.

- The interior point methods predicts more accurately.

Overall, the results suggest that LP relaxation solution can serve as a reasonable prediction to the true solution, with moderate accuracy. This behavior exactly aligns with the case where `PMVB` can be applied.

