# OpenReview forum: "Data-driven Mixed Integer Optimization through Probabilistic Multi-variable Branching"
_ICML.cc/2026/Conference — ICML 2026 regular_

### Official Review · Reviewer_cpm1 · 2026-03-05

**Soundness:** 4
**Presentation:** 3
**Significance:** 3
**Originality:** 3
**Overall Recommendation:** 5
**Confidence:** 3

**Summary:**

The paper proposed a probabilistic framework to perform mixed integer optimization. Compared with existing methods, the proposed method is easy to implement. Moreover, the authors apply classical statistical learning theory to establish a sharp non-asymptotic bound for their prediction. Then the authors provide solutions to some practical issues when implementing this algorithm.

**Compliance With Llm Reviewing Policy:**

Affirmed.

**Key Questions For Authors:**

I think the proposed method heavily hinges on the quality of probability estimate $\tilde{y}_i$'s.
Is there any remedy if the these estimates are not satisfactory?

**Limitations:**

I think the authors have discussed some important limitations of the proposed method.

**Strengths And Weaknesses:**

Overall speaking, the proposed method is easy to implement and has solid theoretical guarantees.
The method is clearly presented.

The paper used concentration equalities to establish theoretical guarantees for the proposed method.
In terms of the presentation, the paper describes the idea of using the predicted probability to generate binary outcome $y_i$'s. But the way how these predicted probabilities are obtained is not clearly provided, which I think has a critical impact on the performance of the performance.

I think the idea of using the predicted probability for binary outcome to generate such data is not a quite novel idea, but applying such an idea to the setting of Integer Optimization leads to an effective and efficient algorithm.

---

> ### Author Rebuttal · Authors · 2026-03-30
>
> **Response to Reviewer cpm1**
>
> Thank you for your efforts in the review process and for your positive evaluation of the paper.
>
> **Weaknesses**
>
> 1. How the predicted probabilities are obtained is not clearly provided.
>
>    Thank you for raising this point. As a key feature of PMVB, it is model-agnostic and is compatible with any mechanism that generates probability predictions. As we demonstrated in the experiments, we can use simple models like logistic regression, advanced models such as GNNs, or predictions from LP relaxation. We will add a more detailed explanation on how to obtain the prediction in the revision. We also kindly refer the reviewer to **Appendix A** of our paper, which includes a summary of existing ML approaches for MIP solving.
>
>
>     We have also conducted additional experiments to further validate the improvement of PMVB with more advanced upstream learning models (see Section A of the https://anonymous.4open.science/r/icml_6391_add_exp-7FAF). We observe that more expressive GNN models improve prediction accuracy, and the speedup achieved by PMVB is positively correlated with this accuracy. This supports the view that PMVB effectively leverages improvements in upstream prediction models.
>
> **Question**
>
> 1. PMVB hinges on the quality of the probability estimate. Is there any remedy if these estimates are not satisfactory?
>
>    This is a very good question. Since PMVB is a data-driven approach, it's inevitable that it will be adversely affected if the upstream prediction model yields misleading results. In such cases, data-driven approaches should generally not be used, and our data-free approach based on LP-relaxation is likely preferred.
>
>    In particular, it is always possible to tighten the LP-relaxations by adding cutting planes. Solvers such as CPLEX provide callback interfaces that allow users to extract the LP relaxation with cutting planes. With these cutting planes, the LP relaxation will be closer to the original MIP problem, and the prediction can be provably improved.
>
> Thank you again for your constructive feedback!

---

> > ### Author Rebuttal · Reviewer_cpm1 · 2026-04-03
> >
> > I think estimating the probability is a critical component of the proposal. If there is no guarantee for this step, a more robust strategy is needed. But the authors do not fully address this concern. Thus I want to change my score to be 4.

---

> > > ### Author Response · Authors · 2026-04-03
> > >
> > > Thank you for your follow-up comments and for your engagement in the review process. We apologize for the confusion in our previous response and would like to clarify the robustness of PMVB more precisely.
> > >
> > > 1. It is generally difficult to obtain theoretical guarantees on probability estimates of data-driven approaches for general MIPs, except under certain assumptions (e.g., i.i.d. settings). In our paper, we provide such guarantees in special cases (e.g., the data-free analysis in Section 4 and the learning-theoretic analysis in Appendix B) to theoretically justify the use of probabilistic estimates in constructing branching hyperplanes.
> > >
> > > 2. More importantly, our theory on data-driven hyperplanes based on Chebyshev inequality does not depend on the quality of estimates to be *correct*; it only depends on the quality of estimates to be *efficient*. When the probability estimates are poor, the accuracy and variance estimates from Section 3.1 will automatically enforce conservative hyperplanes. The theoretical guarantees still hold, but with weaker bounds, leading to reduced efficiency rather than failure. In this sense, the design of PMVB already incorporates robustness.
> > >
> > >    In addition, PMVB can be used as an exact branching rule, so imperfect probability estimates do not affect correctness or optimality; they only influence runtime. It ensures that PMVB remains both **safe and practically robust**, degrading smoothly in the face of low-quality probability estimates.
> > >
> > > We will revise the paper to make this robustness perspective more explicit. We hope this clarification addresses your concern and would appreciate your reconsideration of the evaluation.

---

### Official Review · Reviewer_TS6d · 2026-03-12

**Soundness:** 2
**Presentation:** 3
**Significance:** 2
**Originality:** 2
**Overall Recommendation:** 3
**Confidence:** 4

**Summary:**

The authors propose a new approach for multi-variable branching (MVB) for improving the effectiveness of MILP solvers using ideas from statistical theory. The most appealing feature of the proposal is the simplicity of the approach. It identifies a partition among a group of binary decision variables to set the partitions up for branching up or down. As a result, if this approach yields good results, it can have a positive impact for improving the effectiveness of MILP solves across a large variety of problems. They devise their branching logic based on statistical learning and test their approach and provide results for classical benchmarks and instances arising from real-world problems.

**Compliance With Llm Reviewing Policy:**

Affirmed.

**Key Questions For Authors:**

How does MVB handle context and cost of decisions?
How much of MVB effectiveness is lost if the assumptions in the paper are relaxed or removed?
Can the baselines for comparison be strengthened to reasonably reflect an ordinary practitioner's ability solve a problem that shows repeatedly with different data?

**Limitations:**

yes

**Strengths And Weaknesses:**

The strength of the approach lies in its simplicity and ease of integration into an MIP solve. The idea of risk pooling to groups of binary decision variables is really nice. The focus appears to be on those MIPs that are regularly solved, so one can do statistical learning from offline data to set up the MVB. The assumption of independence among binary variables is very restrictive and difficult to defend, and the authors, to their credit have noted this and they propose a refined alternative. Given that the premise for the branching scheme is statistical learning rather than the structure of MILP and its properties, the assessment of gain comes from empirical performance.

There are few assumptions in the paper
A key weakness is the proposed statistical learning-based branching tends to ignore consequence or context of the training examples. There may be 1000s of training instances in a low-demand environment, but what really matters are those few instances in a high-demand environment, where the branching (e.g. opening or closing of additional warehouse) may be in the entirely opposite direction. This would require the MVB to be contextually aware as well as consequentially aware.

Next, a problem that is solved regularly and with a known pattern of data variations over time, can be rapidly resolved with warm-start and with simple heuristic branching settings based on experience to precisely account for context and consequence of the given next instance. Given this, the comparison of gain has to be with a better baseline that incorporates warm-start and simple heuristic branching settings.

"We assume that the optimal solution map is unique throughout the paper. This assumption can often be satisfied by adding an
arbitrarily small perturbation to the MIP objective."

How critical is this assumption? Real-world MILPs are more likely to come with alternative optimal solutions as well as more near-optimal solutions, which are important when solving NP-Hard problems. What happens if this assumption is relaxed?

---

> ### Author Rebuttal · Authors · 2026-03-30
>
> **Response to Reviewer TS6d**
>
> Thank you for your time in the review process and for your constructive comments.
>
> **Weaknesses**
>
> 1. Contextual awareness
>
>    We agree that, in recurring industrial MIPs, performance depends on whether the method can react to the current instance, rather than only average behavior over historical data. We would like to clarify that PMVB is not itself a forecasting model. PMVB is a downstream procedure that takes as input an instance-specific score or probability vector and converts it into branching hyperplanes. Therefore, contextual awareness is not removed by PMVB; it depends on the upstream model used to generate the probabilities.
>
>    In particular, if the predictor uses features describing the current operating regime, then PMVB uses that regime-specific information as well. Conversely, if the predictor is poorly specified and ignores regime information, any downstream method based on it will suffer. In this sense, PMVB should be viewed as orthogonal to context modeling: it is a mechanism for exploiting instance-level predictions, not a replacement for modeling context.
>
>    To address the case where historical-data-based predictors may be unreliable, this is precisely one motivation for our data-free variant: using the root relaxation as an instance-specific proxy. Since this information is computed from the current instance, it is instance-specific even without a trained model. This makes PMVB useful even when historical patterns are unstable.
>
> 2. Cost/consequence of wrong decisions
>
>    We agree that in many applications, especially when certain decisions are high-impact, one should distinguish between a method used as a heuristic and one used as part of an exact solve.
>
>    This distinction is important for PMVB:
>
>    - As a primal heuristic, PMVB solves the most promising branch, and the goal is to find a strong incumbent early.
>    - As an exact branching rule, PMVB partitions the feasible region into subproblems whose union still covers the original feasible set. Therefore, even if the prediction is poor, PMVB does not compromise optimality; the effect is only on computational runtime.
>
>    So the consequence of inaccurate predictions is not "making the wrong business decision" in the exact setting. Rather, it is potentially losing some runtime improvement. We will make this distinction more explicit in the revision.
>
> 3. Stronger comparison baseline
>
>    Thanks for raising this point, and sorry for the confusion. The baseline solvers actually already use a warm start from the rounded solution. We've now included additional experiments for solvers without warm start. The results are summarized in **Section C** of https://anonymous.4open.science/r/icml_6391_add_exp-7FAF/add_exp.pdf. The observation is that the effect of warm start is solver-specific, but the speedup trend of PMVB is consistent. We also added a comparison with other baseline methods from the literature (**Section B**).
>
> **Questions**
>
> 1. Uniqueness of the MIP solution
>
>    This is a good question. This assumption is made to justify viewing binary variables as Bernoulli random variables (r.v.). If there are multiple solutions where a variable can take 0/1, such behavior can't be fully characterized by r.v.s unless the underlying MIP algorithm is deterministic. In practice, this assumption can be safely removed by assuming that the MIP solver is deterministic (it arrives at the same optimum, even if the optimal solution is not unique), a condition satisfied by most commercial solvers [1]. The theory is essentially the same (i.e., by replacing $y^\star$ by $y^\star_{CPLEX}$). We also note that the idea of using perturbation to avoid degeneracy (multiple solutions) has been used in solvers like IPOPT [2].
>
> 2. How does MVB handle context and the cost of decisions?
>
>    Please see our response to weaknesses.
>
> 3. How much of MVB effectiveness is lost if the assumptions are relaxed?
>
>    As long as the underlying solver is deterministic, our theoretical results in **Section 3** will hold. Since Chebyshev's inequality holds without assuming independence among random variables, it imposes no additional assumptions except requiring bounded variance.
>
>    On the other hand, our data-free results in **Section 4** rely on the problem structure and do not hold for general problems. To empirically validate our approach, we've conducted additional experiments on MIPLIB instances. The experiments show that LP relaxation often provides a moderate-quality prediction of the optimal solutions (~80%). We kindly refer the reviewer to **Section D** of our experiments.
>
> 4. Can the baselines for comparison be strengthened?
>
>    Please see the response to weaknesses.
>
> Thanks again for your time in the review process. We hope that our response addresses your concerns.
>
> **References**
>
> [1] https://support.gurobi.com/hc/en-us/articles/360031636051-Is-Gurobi-deterministic
>
> [2] https://optimization-online.org/2004/03/836/

---

> > ### Author Rebuttal · Reviewer_TS6d · 2026-04-03
> >
> > Gain or value for real-world MILP applications from the purely statistical branching approach proposed rather than fundamental problem structure has not been sufficiently established, especially contextual awareness. Thanks to the authors for their clarification on the baseline.

---

> > > ### Author Response · Authors · 2026-04-04
> > >
> > > Thank you for your follow-up comments and for your engagement in the review process. Below, we directly address your concerns regarding
> > >
> > > 1. context awareness
> > > 2. the gain of PMVB for real-world MILP applications relative to exploiting the fundamental problem structure.
> > >
> > > ---
> > >
> > > **Context-awareness**
> > >
> > > We clarify that PMVB operates on **context-specific inputs**. The role of capturing context-level information lies in the upstream model (e.g., prediction models or LP relaxations), while PMVB converts these predictions into branching decisions. Modern predictors, such as graph neural networks [1, 2], explicitly incorporate both constraint structure and instance data, and are therefore context-aware. For example, [3] enables context-awareness by adopting a tripartite-graph-based representation of MIP instances. [4] adopts a heterograph representation to encode the spatial and temporal context of instances. This graph representation and context-awareness are further investigated in [5].
> > >
> > > In summary, context-awareness can be achieved using advanced learning models. However, to our knowledge, no existing baseline based on context-aware predictions works without deep integration with the solver's internal behavior.
> > >
> > > ---
> > >
> > > **Gain over exploiting problem structure**
> > >
> > > We fully agree that exploiting problem structure is essential for real-world MIP solving. Our intention is not to replace structure-aware approaches, but to position PMVB as a **complementary mechanism**. In particular, we are not suggesting that users should abandon structure-exploiting methods; rather, PMVB can be used alongside existing pipelines with minimal modification. Its simplicity allows it to integrate naturally with solver-based heuristics and decomposition techniques.
> > >
> > > Beyond this, PMVB can provide additional performance gains in settings where structure is difficult to identify or exploit [6]. In many industrial applications, models are highly complex, and developing effective structure-based methods can require substantial domain expertise and engineering effort. In such cases, PMVB can be deployed immediately as a flexible and practical enhancement while remaining compatible with future structure-based improvements.
> > >
> > > ---
> > >
> > > We will revise the paper to make this positioning clear:
> > >
> > > 1. When a context-aware upstream prediction model is used (e.g., recent GNN architectures), PMVB is indeed context-aware.
> > > 2. PMVB complements structural methods by incorporating additional instance-specific information, rather than replacing them.
> > >
> > > We hope this clarification addresses your concerns and would appreciate your reconsideration of the evaluation.
> > >
> > > ---
> > >
> > >
> > > **References**
> > >
> > > [1] Khalil, E. B., Morris, C., & Lodi, A. (2022, June). Mip-gnn: A data-driven framework for guiding combinatorial solvers. In *Proceedings of the AAAI Conference on Artificial Intelligence*(Vol. 36, No. 9, pp. 10219-10227).
> > >
> > > [2] Gasse, M., Chételat, D., Ferroni, N., Charlin, L., & Lodi, A. (2019). Exact combinatorial optimization with graph convolutional neural networks. *Advances in neural information processing systems*, *32*.
> > >
> > > [3] Li, S., Ouyang, W., Paulus, M., & Wu, C. (2023). Learning to configure separators in branch-and-cut. Advances in Neural Information Processing Systems, 36, 60021-60034.
> > >
> > > [4] Jung, H., Park, J., & Park, J. (2022). Learning context-aware adaptive solvers to accelerate quadratic programming. arXiv preprint arXiv:2211.12443.
> > >
> > > [5] Wu, C., Chen, Q., Wang, A., Ding, T., Sun, R., Yang, W., & Shi, Q. (2024). On representing convex quadratically constrained quadratic programs via graph neural networks. arXiv preprint arXiv:2411.13805.
> > >
> > > [6] Hooker, J. N. (2019). Logic-based Benders decomposition for large-scale optimization. In Large scale optimization in supply chains and smart manufacturing: Theory and applications (pp. 1-26). Cham: Springer International Publishing.

---

### Official Review · Reviewer_NRdS · 2026-03-12

**Soundness:** 3
**Presentation:** 3
**Significance:** 3
**Originality:** 2
**Overall Recommendation:** 5
**Confidence:** 4

**Summary:**

Mixed-integer programming (MIP) problems are often time-consuming to solve, and in some industrial settings, similar MIP instances need to be solved repeatedly on a daily basis. This makes it natural to ask how past data can be used to speed up future solving, which is exactly where machine learning can help. To address this, the paper proposes Probabilistic Multi-Variable Branching (PMVB), a special type of branching rule. The method uses probability predictions for variables and a statistically motivated construction of branching hyperplanes based on past data. Since the method still works by partitioning the feasible region, it can be used either as a heuristic to explore the region that is most likely to contain the optimum, or as an exact external branching scheme that continues searching over the induced subregions. PMVB can be easily combined with any machine learning model that outputs variable probabilities, and unlike many learning-based methods for branch-and-cut, it does not require modifying the internals of the solver, which is especially practical for commercial black-box solvers. The paper also proposes a data-free version of PMVB, where the fractional root LP relaxation solution is directly used as a proxy for variable probabilities. Experiments on several datasets show some gains over the baselines.

**Compliance With Llm Reviewing Policy:**

Affirmed.

**Final Justification:**

The authors have addressed most of my concerns during the rebuttal period. I believe this paper will add value to the field of machine learning for combinatorial optimization. I recommend accepting this paper.

**Key Questions For Authors:**

1. In the proof of Theorem 4.1, the authors write that "another inequality w.r.t. $\mathcal{L}$ trivially holds." I can follow the argument for the $U$-side inequality, but for the $\mathcal{L}$-side inequality, I do not see how to obtain the same constant $4\sqrt{2}$. I can only see how to recover the same order $O(n^{3/4})$. Could the authors clarify or complete this part of the argument?
2. Since the data-free version uses the root LP relaxation solution as a proxy for binary probabilities, do the authors have additional evidence that this proxy is reliable beyond the randomized knapsack setting analyzed in the paper?

**Minor Comment**
1. Page 3, Line 129: should $\hat{y}_i$ be defined as $\lfloor \tilde{y}_i + \frac{1}{2} \rfloor$ rather than $\lceil \tilde{y}_i + \frac{1}{2} \rceil$?

**Limitations:**

Yes.

**Strengths And Weaknesses:**

**Strengths**
1. This paper is well-written and the presentation is clear. The paper is also structured in a natural step-by-step way, starting from an idealized model and theoretical motivation, and then moving to more practical settings. This helps the reader understand both the motivation and the design choices behind the method.
2. The proposed method is simple and effective, and it can be combined with many machine learning models or black-box commercial solvers. This makes the method flexible and easy to plug in. It can be used both as a heuristic and as a branching rule. In particular, when used as a heuristic, this simple method can provide useful early incumbent solutions without changing the solver internals much, which is very practical.
3. The method is theoretically grounded, and the method itself is motivated from the theoretical side. This makes the overall idea quite natural and helps justify the design of the method and its hyperparameters.

**Weaknesses**
1. The data-free version seems somewhat limited, both theoretically and empirically. On the theory side, it is analyzed only for a simple single-knapsack setting under strong assumptions. On the experimental side, its performance also appears more limited and sensitive. This may be related to the fact that directly treating the LP solution as a probability prediction for the MIP solution can be quite inaccurate. A coordinate of the LP solution being close to $1$ can still most likely correspond to $0$ in the optimal MIP solution for some problems.
2. The experimental baselines also seem somewhat limited. The paper mainly compares against commercial solvers directly. However, those solvers do not use any historical data, while PMVB does. So at least some comparisons against other data-aware or learning-based approaches would make the empirical evaluation more convincing.

---

> ### Author Rebuttal · Authors · 2026-03-30
>
> **Response to Reviewer NRdS**
>
> Thank you for your time in the review process and for your constructive comments.
>
> **Weakness**
>
> 1. Data-free variant is limited; experiments are sensitive.
>
>    Theoretically, we first wish to note that establishing theoretical guarantees for learning-based approaches to general MIP solving is challenging, and even our current results are new in the literature. To address your concerns, we present a multi-knapsack (MKP) extension of our theory, which we'll include in the revision. This extension also holds for covering-type constraints. We summarize the main idea: consider the MKP:
>
>    $$
>    \max_{y\in\{0,1\}^n} \sum_{i=1}^n c_i y_i ~~s.t. ~~\sum_{i=1}^n a_i y_i\le b,~a_i\in {R}_+^m,
>    $$
>    Let $\tilde{y}$ be an optimal vertex solution, and define
>
>    $$U=\\{i:\tilde y_i=1\\},\quad L=\\{i:\tilde y_i=0\\},\quad F=\\{i:0<\tilde y_i<1\\}.$$
>
>    Since the problem has $m$ knapsack constraints, any vertex solution has at most $m$ fractional variables, hence $|F|\le m$. Introduce dual prices $\lambda^\*$ and reduced costs
>
>     $$\Delta_i:=c_i-(\lambda^\*)^\top a_i.$$
>
>    Assuming that $(c, a_i)$ are i.i.d. , we can show the following anti-concentration property $\mathbb{P}\\{|\Delta_i|\le\varepsilon|\lambda^\*\\}\le\kappa\varepsilon,$
>
>    for some constant $\kappa > 0$. Just like the single-constraint case, it is expected that the integer optimum agrees with the LP classification away from the boundary region $\\{i:|\Delta_i|\le \varepsilon\\}$, so we obtain
>
>    $$\sum_{i\in U}y_i^\*\ge |U|-m-O(\kappa\varepsilon n+\sqrt{n\log(1/\eta)})$$ and
>    $$\sum_{i\in L}y_i^\*\le m+O(\kappa\varepsilon n+\sqrt{n\log(1/\eta)}).$$
>
>    The prediction error has three terms:
>
>    - the # of LP-fractional variables $|F|\le m$;
>    - width of the reduced-cost boundary region $O(\kappa \varepsilon n)$;
>    - a noise term of $O(\sqrt{n\log(1/\eta)})$.
>
>    and we can derive hyperplanes as in the single-constraint case.
>
>    ---
>    For your concerns on the experimental side of using LP relaxation, we've added empirical studies on MIPLIB instances to investigate whether LP solution helps predict the MIP solution (see response to questions). We also remark that in practice, our data-free approach can be enhanced by resorting to *root* relaxations with instance-wise cutting planes. Root relaxations can be much tighter than the LP relaxation, and solvers such as CPLEX allow users to access them through callbacks.
>
> 2. Limited baselines
>
>    Thanks for the comments and sorry for the confusion. Our experiments already let solvers use warm start, and we will clarify in the revision. We have included two additional experiments, including
>
>    - New baseline from literature
>
>      We compare PMVB to the advanced Predict-and-Search approach [1] from literature on two NeurIPS benchmarks, and PMVB achieves better performance on one, while being comparable on the other.
>
>    - Solver without warm start from rounded predicton.
>
>      The results show that the effect of warm start is solver and instance-specific, but the speedup trend remains consistent.
>
>    Please see **Section B&C** of https://anonymous.4open.science/r/icml_6391_add_exp-7FAF/ for details.
>
> **Questions**
>
> 1. Theorem 4.1
>
>    Thanks for catching the issue. As you noticed, the proof of the $L$-side inequality in Thm 4.1 does not follow automatically, and the constant is indeed different. The proof structure is the same: first, according to (21) and the inequality at the bottom of page 15,
>
>    $$\sum_{i\in U^\*\setminus U}a_i \le 1+\sum_{i\in U\setminus U^\*}a_i\leq1+4\sqrt{n}.$$
>
>    Then, let $S = U^\* \setminus U$, if $|S| \geq 4\sqrt{2}n^{\frac{3}{4}} + n^{\frac{1}{4}}$, by Lemma C.4 with $\delta = \frac{1}{2\sqrt{n}}$,
>
>    $$\sum_{i\in U^\*\setminus U}a_i\ge\frac{|S|^2}{8n}\ge\frac{(4\sqrt{2}n^{\frac{3}{4}}+n^{\frac{1}{4}})^2}{8n}\ge 4\sqrt{n}+\sqrt{2},$$
>
>    which contradicts with the above inequality. Thus, $|U^\* \setminus U| \leq 4\sqrt{2}n^{\frac{3}{4}} + n^{\frac{1}{4}}.$ Besides, by definition, $L\cap U^\* \subseteq ([n] \setminus U) \cap U^\* = U^\*\setminus U,$  and
>
>    $$\sum_{i\in L}y_i^\*=|L\cap U^\*| \le |U^\*\setminus U| \le 4\sqrt{2}n^{3/4}+n^{1/4}.$$
>
>     Thus the same $O(n^{3/4})$ order follows. We'll revise the theorem and proof.
>
> 2. Evidence that this proxy is reliable
>
>    This is a very good question. To address the question, we've conducted additional experiments to investigate the reliablity of LP relaxation as a proxy on MIPLIB instances. In summary, on the tested instances, LP relaxation can often achieve an accuracy > 80%, which is sufficient to justify using PMVB in the data-free setting. We kindly refer the reviewer to **Section D** of our additional experiments.
>
> 3. Minor comment.
>
>     Yes, it should round down. Thanks for catching it!
>
> We hope that our responses address your concerns. Thanks again for your efforts in the review process!
>
> **References**
>
> [1] Han, Q., et. al. A GNN-guided predict-and-search framework for MILP.

---

> > ### Author Rebuttal · Reviewer_NRdS · 2026-04-04
> >
> > Thank you for the clarifications, the theoretical results now make sense. I will keep my positive score.

---

> > > ### Author Response · Authors · 2026-04-04
> > >
> > > Thank you for your thoughtful feedback and for confirming that your concerns have been fully addressed. We truly appreciate your time and engagement in the review process.
> > >
> > > If you feel that the clarifications and additional experiments strengthen the contribution, we would be grateful if you could consider raising the score.
> > >
> > > Thank you again for your support.

---

### Official Review · Reviewer_tCTB · 2026-03-13

**Soundness:** 4
**Presentation:** 3
**Significance:** 2
**Originality:** 3
**Overall Recommendation:** 4
**Confidence:** 4

**Summary:**

This paper introduces PMVB, a data-driven method designed to accelerate the solving of MILP problems using machine learning models. The approach first predicts the probability distribution of binary variables using models such as GNNs, and then leverages concentration inequalities from statistical learning theory to construct data-driven branching hyperplanes that partition the feasible region into disjoint sub-regions for solving. The paper not only demonstrates the highly minimalist code implementation of this method but also provides a detailed theoretical basis for hyperparameter selection using statistical bounding theory.

**Compliance With Llm Reviewing Policy:**

Affirmed.

**Final Justification:**

My questions have been fully resolved, and I will maintain my positive score.

**Key Questions For Authors:**

I have two main questions that I hope the authors can address in their rebuttal, which will help me finalize my assessment of the paper's technical depth and innovativeness. First, regarding the choice of model architecture, many new GNN architectures (such as networks with Attention mechanisms) or architectures specifically designed for integer outputs have emerged recently to better learn solution distributions; why did the authors choose the current, relatively outdated network structure instead of adopting these newer, more expressive architectures, and could you discuss the application potential or provide comparative experiments using new architectures to help evaluate the performance upper bound of this framework? Second, considering that the overall idea of predicting and adding branching hyperplanes is very similar to the Predict-and-Search framework (Han et al., 2023)—since restricting the search radius in P&S is essentially also a cut—could the authors further elaborate on the core differences between PMVB and P&S in terms of algorithmic mechanisms and practical applications, aside from the excellent mathematical bounding proofs already provided, as clarifying this will directly impact my judgment of the paper's originality?

**Limitations:**

yes

**Strengths And Weaknesses:**

The paper excels in its significance and theoretical soundness, as the authors provide highly rigorous mathematical bounding proofs and leverage statistical learning theory to establish a solid theoretical foundation for constructing branching hyperplanes, which makes a positive contribution to the advancement of data-driven MIP research and serves as the core reason for my overall positive evaluation. However, there are certain flaws in the originality and the overall refinement of the methodology. First, the overall concept is quite similar to the existing Predict-and-Search framework (e.g., Han et al., 2023), which essentially also adds a form of cut by predicting and restricting the search radius to not exceed K, making the core motivation of this paper somewhat less novel. Second, the specific technical implementation appears slightly unrefined, as the network architecture used to learn the solution distribution is relatively outdated and fails to fully incorporate the latest advancements in graph neural network architecture design within the deep learning community.

[1Han Q, Yang L, Chen Q, et al. A GNN-Guided Predict-and-Search Framework for Mixed-Integer Linear Programming[C]. The Eleventh International Conference on Learning Representations.]

---

> ### Author Rebuttal · Authors · 2026-03-30
>
> **Response to Reviewer tCTB**
>
> Thank you for your efforts in the review process and for acknowledging the theoretical soundness of our results.
>
> **Weaknesses**
>
> 1. Flaws in the originality; similarity to Predict-and-Search (P&S).
>
>    We appreciate this important point and would like to clarify the distinction from two perspectives.
>
>    **Theoretical perspective**. As noted in your review, a key strength of PMVB is its theoretical grounding. Designing ML+MIP approaches beyond heuristics is challenging due to the discrete and highly general nature of MIP problems. In particular, hyperparameter tuning in existing approaches often relies on empirical trial and error.  In contrast, PMVB provides a principled, learning-theoretic framework for parameter selection, providing interpretable and theoretically justified hyperparameter selection. We believe this perspective is broadly applicable and can inform the design of future ML+MIP methods, including existing frameworks such as P&S.
>
>    **Comparison with P&S**
>
>    While PMVB shares high-level similarities with P&S [1], there are several important conceptual and practical differences.
>
>    PMVB partitions variables into two sets \(U\) and \(L\) based on rounded predictions and constructs two hyperplanes:
>    $$
>    \sum_{j \in L} y_j\le\xi_L\Leftrightarrow\\|\mathbf{y}_L-\hat{\mathbf{y}}_L\\|_1\le\xi_L,~\text{and}~~\\|\mathbf{y}_U-\hat{\mathbf{y}}_U\\|_1\le\xi_U,
>    $$
>    where $\xi$ are chosen by the concentration inequalities. In contrast, P&S enforces a joint trust-region constraint:
>    $$
>    \\|\mathbf{y}_L-\hat{\mathbf{y}}_L\\|_1+\\|\mathbf{y}_U-\hat{\mathbf{y}}_U\\|_1 \leq \Delta
>    $$
>    and $\Delta$ is a trust-region radius selected heuristically. There are several key differences:
>
>    - *Parameter selection*
>
>      PMVB comes with a parameter tuning strategy motivated from theory, while the parameter in P&S needs manual tuning. In addition, our theory can also be generalized to guide parameter selection in P&S.
>
>    - *Variable partitioning*
>
>      Separating variables into $U$ and $L$ allows PMVB to exploit asymmetry in prediction quality. For example, in problems with partitioning constraints ($\sum_j y_j=1$), most variables are zero in the optimal solution, making predictions in $U$ more informative. A joint constraint (as in P&S) mixes these predictions and can weaken the resulting cuts. Our additional experiments also confirm this effect (see the second additional experiment).
>
>    - *Role in optimization*
>
>      P&S is a primarily primal heuristic, while PMVB can be used to solve an MIP to provable optimality.
>
>    We have also included new experiments comparing PMVB and P&S under the same probabilistic predictions, isolating the effect of the branching strategy.
>
> 2. Technical implementation appears slightly unrefined.
>
>    We appreciate this comment and would like to clarify our design choice. PMVB is intentionally model-agnostic, as we aim to decouple:
>
>    - how predictions are generated (upstream model), and
>    - how predictions are used (PMVB).
>
>    This design makes PMVB compatible with a wide range of models. In our experiments, we include both simple models (logistic regression) and recent GNN architectures tailored for MIP [2, 3], demonstrating this flexibility. But we also agree that the choice of upstream model is important in practice. To address this, we've added additional experiments (see below) to evaluate more advanced architectures. We will also include a more comprehensive comparison in the revision.
>
> **Additional experiments**
>
> Per the reviewer's suggestion, we conducted additional experiments (**Section A/B** of the https://anonymous.4open.science/r/icml_6391_add_exp-7FAF).
>
> 1. Advanced GNN architectures
>
>    We observe that more expressive GNN models improve prediction accuracy, and the speedup achieved by PMVB is positively correlated with this accuracy. This supports the view that PMVB effectively leverages improvements in upstream prediction models.
>
> 2. Comparison with P&S
>
>    We evaluate both methods on two NeurIPS 2021 benchmarks: load balancing (LB) and item placement (IP), using the same probabilistic predictions.
>
>    - On LB, PMVB outperforms P&S. LB instances have partitioning-type constraints, and separating $U$ and $L$ is particularly beneficial.
>    - On IP, PMVB achieves performance comparable to P&S.
>
> **Questions**
>
> 1. Choice of GNN architecture
>
>    We have added experiments with more advanced architectures (see above), showing consistent improvements in prediction quality and corresponding gains in PMVB performance.
>
> 2. Difference between PMVB and P&S
>
>    Please see our response to the weaknesses.
>
> Thanks again for your constructive feedback!
>
> **References**
>
> [1] Han et al. (2023). A GNN-guided predict-and-search framework for MILP.
>
> [2] Khalil et al. (2022). MIP-GNN.
>
> [3] Cantürk et al. (2024). Scalable primal heuristics with GNNs.

---

> > ### Author Rebuttal · Reviewer_tCTB · 2026-04-03
> >
> > Thanks for the detailed rebuttal. You have clearly explained the algorithmic and practical differences between PMVB and the Predict-and-Search framework. I also greatly appreciate the new comparative experiments using more advanced GNN architectures, which effectively address my concerns about the model's performance upper bound. My questions have been fully resolved, and I will maintain my positive score.

---

> > > ### Author Response · Authors · 2026-04-04
> > >
> > > Thank you for your thoughtful feedback and for confirming that your concerns have been fully addressed. We truly appreciate your time and engagement in the review process.
> > >
> > > If you feel that the clarifications and additional experiments strengthen the contribution, we would be grateful if you could consider raising the score.
> > >
> > > Thank you again for your support.

---

### Decision · Program_Chairs · 2026-04-30

**Decision:**

Accept (regular)

**Comment:**

It proposes a novel multi-variable branching (MVB) approach to improve the effectiveness of MILP solvers, supported by bounds from statistical theory. The implementation is simple, effective, and broadly applicable. Two reviewers have no remaining concerns after the rebuttal and strongly support accepting the work. From my understanding, the authors have clarified the remaining concerns raised by the other two reviewers regarding guarantees of probabilistic estimation, context awareness, and exploitation of problem structure. Thus, I am inclined to accept the work.